# Focal amplifications are associated with chromothripsis events and diverse prognoses in gastric cardia adenocarcinoma

Xue-Ke Zhao[1,3], Pengwei Xing[2,3], Xin Song[1,3], Miao Zhao[2], Linxuan Zhao [2], Yonglong Dang[2], Ling-Ling Lei[1], Rui-Hua Xu[1], Wen-Li Han[1], Pan-Pan Wang[1], Miao-Miao Yang[1], Jing-Feng Hu[1], Kan Zhong[1], Fu-You Zhou[1], Xue-Na Han[1], Chao-Long Meng[1], Jia-Jia Ji[1], Xingqi Chen [2,4✉] & Li-Dong Wang [1,4✉]

The role of focal amplifications and extrachromosomal DNA (ecDNA) is unknown in gastric cardia adenocarcinoma (GCA). Here, we identify frequent focal amplifications and ecDNAs in Chinese GCA patient samples, and find focal amplifications in the GCA cohort are associated with the chromothripsis process and may be induced by accumulated DNA damage due to local dietary habits. We observe diverse correlations between the presence of oncogene focal amplifications and prognosis, where *ERBB2* focal amplifications positively correlate with prognosis and *EGFR* focal amplifications negatively correlate with prognosis. Large-scale ERBB2 immunohistochemistry results from 1668 GCA patients show survival probability of ERBB2 positive patients is lower than that of ERBB2 negative patients when their surviving time is under 2 years, however, the tendency is opposite when their surviving time is longer than 2 years. Our observations indicate that the *ERBB2* focal amplifications may represent a good prognostic marker in GCA patients.

[1] State Key Laboratory of Esophageal Cancer Prevention & Treatment and Henan Key, Laboratory for Esophageal Cancer Research of The First Affiliated Hospital, Zhengzhou University, 450052 Zhengzhou, Henan, PR China. [2] Department of Immunology, Genetics and Pathology, Uppsala University, 75108 Uppsala, Sweden. [3] These authors contributed equally: Xue-Ke Zhao, Pengwei Xing, Xin Song. [4] These authors jointly supervised this work: Xingqi Chen, Li-Dong Wang. ✉email: xingqi.chen@igp.uu.se; ldwang2007@126.com

Extrachromosomal DNA (ecDNA) was first identified more than half a century ago[1], and has been associated with genomic instability[2,3]. With next-generation sequencing technologies and high throughput imaging platforms, an increasing number of studies have shown that ecDNAs are present in most tissues, and contribute to the intratumoral heterogeneity and cancer progression[2,4–9]. Using computational analysis of whole-genome sequencing (WGS) data from a large-scale cancer cohort, it has been demonstrated that the presence of ecDNA is cancer-type specific, and is associated with oncogene amplification and poor outcomes across multiple cancers[7]. Focal amplifications in cancer often involve the juxtaposition of rearranged segments of DNA from distinct chromosomal loci into a single amplified region[9–16], and focal amplifications in nearly half of the samples across a variety of cancer types can be explained by ecDNA formation[12,17]. ecDNA was also proposed as the primary driver of focal amplifications, enabling oncogene amplifications and rapid tumor evolution[9]. Thus, it is very valuable to understand the functions of ecDNA in tumors by exploring focal amplifications in clinical samples. The cardia is located between the esophagus and the stomach. Gastric cardia adenocarcinoma (GCA) and esophageal squamous cell carcinoma (ESCC) occur together in the Taihang Mountains of north central China at high rates[18–20]. Gastric cancer in this area occurs primarily in the uppermost portion of the stomach and is referred to as GCA, and those in the remainder of the stomach are called gastric noncardia adenocarcinoma (GNCA)[21]. Adenocarcinomas from junction of esophagogastric junction are usually classified as Siewert type II of esophagogastric junction adenocarcinoma in western countries[22–26], where Barrett's esophagus is very common and has been considered as an important precancerous lesion of adenocarcinoma at esophagogastric junction[27]. However, GCA from a Chinese population in this area has distinct features compared to Western countries[20,27,28], and very low frequency of Barrett's esophagus is observed[27]. Instead, GCA in this area shares similar features with that of esophageal squamous cell carcinoma[20,27]. A previous study reported that oncogene amplification and gene rearrangements drive the progression and poor prognosis of GCA[29]. However, it is still unclear whether focal amplifications and ecDNA is present in GCA, and what role they play in the GCA progression or whether it is correlated with patient prognosis.

In this work, we investigate the availability and function of focal amplification and ecDNA in GCA in a Chinese cohort of GCA using whole-genome sequencing (WGS), whole-exome sequencing (WES), and immunohistochemistry, and explore the relationship between the presence of oncogene focal amplifications and prognosis in GCA. We identify the focal amplifications and ecDNA amplicons present in most GCA patients, and find focal amplifications in the GCA cohort are associated with the chromothripsis process and may be induced by accumulated DNA damage due to local dietary habits. We observe diverse correlations between the presence of oncogene focal amplifications and prognosis. Large-scale *ERBB2* immunohistochemistry results from 1668 GCA patients show survival probability of ERBB2 positive patients is lower than that of *ERBB2* negative patients when their surviving time is under 2 years, however, the tendency is opposite when their surviving time is longer than 2 years. Our observations indicate that the *ERBB2* focal amplifications may represent a good prognostic marker in GCA patients.

## Results

### Characterization of focal amplifications and ecDNA amplicons in GCA.
Since focal amplifications and ecDNA can be identified from WGS data using amplification region reconstruction tool, AmpliconArchitect (AA)[2,4–7,9,30], we first performed WGS of 36 pairs of GCA tumor and tumor-adjacent normal tissue from a high incidence GCA rate region in the northern region of China, Henan Province (see "Methods" section). All of our WGS data in 36 pairs of samples had sufficient sequencing coverage and a high mapping rate (>95% mapping rate) (Supplementary Fig. 1a and Supplementary Data 1). In addition, we performed single-nucleotide variant (SNV) analysis in the 36 GCA patients and found that the top ranking mutated cancer driver-genes[31–33] (81% mutation rate) was TP53 (Supplementary Fig. 1b), which agrees with previous gene mutation studies in GCA patients[21,27,29,34]. Then, we applied AA to these 36 pairs of whole genome sequencing (WGS) data pertaining to GCA tumor and tumor-adjacent normal tissue (Fig. 1a). Following the AA pipeline, we treated the tumor-adjacent normal tissue as the background to call the somatic copy number alteration (CNA) and identified focal amplifications in our GCA cohort. Using this strategy, focal amplifications were identified in 28 of 36 GCA patients (Fig. 1b), and the frequency (77.8%) of focal amplifications observed in our GCA cohort is similar to that of esophageal cancer (~80%) but higher than that of gastric cancer (~50%) in a previous report[7]. Moreover, the number of focal amplifications identified from individual patients showed the high heterogeneity across the GCA cohort (Fig. 1b), with a range of focal amplifications from 0 to 24. For most patients, the number of focal amplifications was less than 10, and only five patients had more than ten focal amplifications (Fig. 1b). In our GCA cohort, focal amplifications were further classified into five categories[7] (Fig. 1b and Supplementary Fig. 1c–e and Supplementary Data 2): circular ($n = 45$) (ecDNA), complex ($n = 21$), linear ($n = 50$), breakage-fusion-bridge (BFB) ($n = 4$), and invalid ($n = 31$), which occurred heterogeneously across the GCA patient cohort (Fig. 1b). We further validated the circular feature of circular focal amplifications identified from AA software using another in silico method, Circle-finder, which identifies circular DNA from paired-end high-throughput sequencing data[35–37]. By checking the sequencing read orientation and junction points of circular focal amplifications using Circle-finder, we found that 89.94–100% of circular focal amplifications identified from AA contained the same junctional reads detected by Circle-finder (Supplementary Fig. 1f–h). The high proportion of overlapping circular focal amplifications from Circle-finder and AA results convinced us that the circular focal amplifications identified with AA are reliable.

Next, we analyzed the size of focal amplifications in our GCA cohort. The size of focal amplifications from GCA ranged from 100 Kbp to 22.6 Mbp, with a median size of 350 Kb (Supplementary Fig. 2a), where 75% of focal amplifications were between 1–2 Mbp, and only 1% of focal amplifications were larger than 20 Mbp (Supplementary Fig. 2b). Some large focal amplifications (>20 Mbp) could be deconvoluted into multiple potential combinations of amplicons using AA software (Supplementary Fig. 2c). Since deconvolution is performed using a computational prediction, there is still the possibility that multiple structures from these large focal amplifications are independent from circular amplicons. We also investigated the frequency of focal amplifications in different chromosomes. We found focal amplifications of different lengths in all chromosomes (Supplementary Fig. 2d, e) and the number of focal amplifications in the different chromosomes was independent of the length of the chromosome (Supplementary Fig. 2d). We concluded that focal amplifications occur heterogeneously across GCA patients (Fig. 1b and Supplementary Fig. 2e).

Next, we performed genomic annotation for all focal amplifications (Fig. 1c and Supplementary Fig. 2f, h). We found that focal amplifications occurred in different parts of the

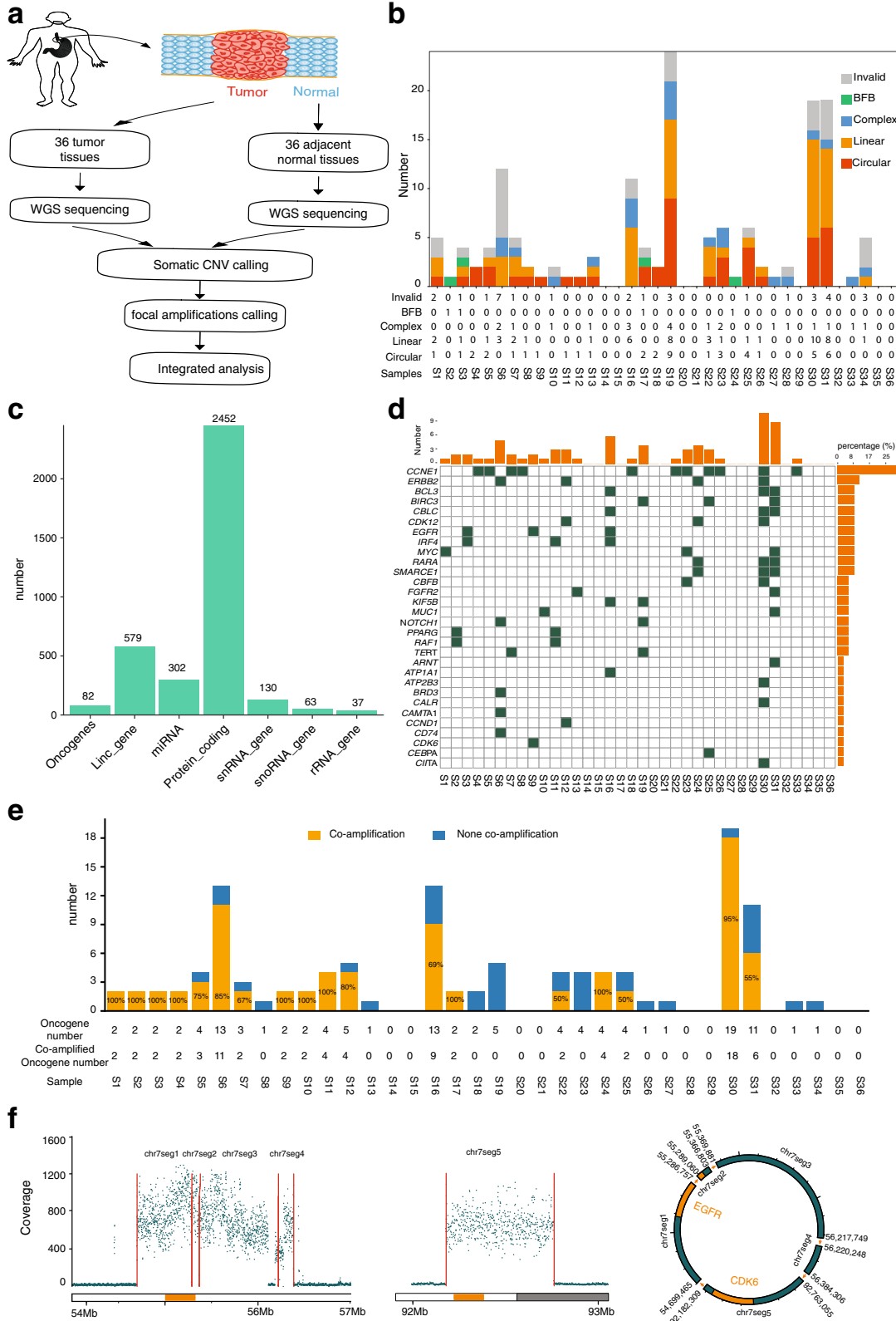

genome, including 2452 sites in protein coding regions and 579 sites in long intergenic non-protein coding RNA (lincRNA) (Fig. 1c). However, the frequency of focal amplifications observed in coding regions (6.28%) was higher than the proportion of coding regions in the whole genome (3.48%) (Supplementary Fig. 2f). Furthermore, the proportion of focal amplifications detected in the exons (14.5%) is higher than that of exons in the

entire genome (9.2%) (Supplementary Fig. 2g). These focal amplifications are also identified at regions of small RNAs (Fig. 1c), including miRNAs (302 sites), SnRNAs (130 sites), SnoRNAs (63 sites), and rRNAs (37 sites). Interestingly, we found that 82 focal amplifications containing canonical oncogenes (CDK12 was reported as a tumor suppressor gene but with oncogenic properties[38]) (Fig. 1c). Next, we focused on the

**Fig. 1 Identification and characterization of focal amplifications from whole-genome sequencing data of the GCA cohort. a** Schematic of the experiment design for detecting focal amplifications from WGS data of 36 pairs of GCA tumor and tumor-adjacent normal tissue from a high incidence GCA rate region in the northern region of China. **b** Detailed characterization of focal amplifications from 36 GCAs, where ecDNA amplicons are further classified into circular (ecDNA), complex, linear, breakage-fusion-bridge (BFB), and invalid. **c** Genomic annotation of all focal amplifications, where the annotation was defined by overlapping gene regions and regions of focal amplifications. **d** Distribution of high-frequency oncogene focal amplifications across all 36 samples. **e** The summary of oncogene focal amplifications co-amplification in our cohort, where co-amplification is defined when two or more than two oncogenes are in the same focal amplifications; **f** *EGFR* and *CDK6* are located in the same circular focal amplification (ecDNA), where the genome coverage on the left panel represents gene amplification of *EGFR* and *CDK6*, and the circular structure on the right panel is the reconstruction of *EGFR* and *CDK6* in the same ecDNA. Source data are provided as a Source Data file for Fig. 1b–f.

analysis of focal amplifications containing oncogenes in our GCA cohort (Fig. 1d). The oncogene focal amplifications across the GCA cohort exhibited a high heterogeneity, and the number of such oncogene focal amplifications varied from 1 to 11 (Fig. 1d, e). Amplification of the cyclin-E1 (*CCNE1*) in the GCA was observed in a previous report[39]. Specifically, we found that *CCNE1* focal amplifications occurred in 11 patients in our cohort (Fig. 1d). *ERBB2* is a member of the human epidermal growth factor receptor (*EGF* family), and it has been reported that *ERBB2* amplification plays an important role in GCA progression[39]. We found that four patients had *ERBB2* focal amplifications (Fig. 1d). The, *CDK12* (also a tumor suppressor gene), *EGFR* and *MYC*, oncogeneswere also found in the focal amplifications format in more than three patients in the cohort (Fig. 1d). The other name for *ERBB2* is *HER2*, and *EGFR* is also called *HER1* or *ERBB1*[40]. Both *HER1* and *HER2* are members of the *EGF* family. The identification of *HER1* focal amplifications and *HER2* focal amplifications in GCA reflects the role of the EGF family in GCA progression[41]. However, we did not observe codetection of *HER1* focal amplifications and *HER2* focal amplifications in the same GCA patient (Fig. 1d), which likely indicates the heterogeneous features in our GCA cohort. The frequent detection of focal amplifications in The Cancer Genome Atlas (TCGA) reflects the presence of cancer specific oncogene focal amplifications in each cancer type[7], where the focal amplifications from gastric cancer and esophagus cancer are investigated. Since the cardia is located at the junction of esophageal and stomach, we next investigated whether the list of oncogene focal amplifications from GCA was similar to that of gastric cancer or esophageal cancer using the TCGA report. We found that GCA shares some common oncogene focal amplifications with both gastric cancer and esophageal cancer including *CCNE1*, *EGFR*, and *MYC* (Supplementary Fig. 3). The top two ranking oncogene focal amplifications, *ERBB2* and *CCNE1*, were the same in both gastric cancer and GCAs. However, the top ranking list of oncogene focal amplifications was different between esophageal cancer and GCAs (Supplementary Fig. 3), where *CCND1* and *EGFR* were the top two ranking oncogene focal amplifications in the esophageal cancer. Our results indicate that the top oncogene focal amplifications from GCAs is more similar to those from gastric cancer. In addition, we observed that several oncogenes oncogene focal amplifications appear in the same GCA patient (Fig. 1d). The cyclization of oncogene focal amplifications is highly amplified due to its rolling-circle replication mechanism, and the circular focal amplifications could contain different oncogenes from different regions of the genome[2]. Thus, we examined whether these different oncogenes in the same patient were located in the same focal amplifications. We first divided the highly amplified regions into segments, recombined them together by read orientation and read junctions, and further reconstructed circular ecDNA containing multiple oncogenes focal amplifications (Fig. 1d–f and Supplementary Fig. 4a–d and Supplementary Data 3). We referred multiple (two or more than two) oncogenes in the same focal amplifications as oncogene focal

amplifications co-amplification (Fig. 1d), and investigated the frequency of such occurrences (Fig. 1d, e). We found i) co-amplification of oncogenes occurred in 50% of patients (18 of 36 patients) (Fig. 1e and Supplementary Fig. 4a); ii) the frequency of oncogene co-amplification varied from 50 to 100% of all oncogene amplifications in different patients (Fig. 1e); and iii) some pairs of oncogene co-amplifications were observed in more than one patient (Supplementary Data 3), where oncogene focal amplifications pairs of *ERBB2* and *CDK12*, *RARA*, and *SMARCE1*, and *CBLC* and *BCL3* occurred in three patients; oncogene focal amplifications pairs of *EGFR* and *IRF4*, *PPARG*, and *RAF1*; and pairs of *CDK12*, *ERBB2* and *RARA* occurred in two patients. Interestingly, *EGFR* and *CDK6* with a physical distance of 40 Mbp, are located in the same circular focal amplifications (Fig. 1f). Using the normal genome copy number as the background, we found that the *EGFR* and *CDK6* circular focal amplifications were amplified forty times compared to other parts of the genome (Fig. 1f). The coamplification of *EGFR* and *CDK6* in the same circular focal amplifications indicates that different genes could work together during the progression of GCAs.

**Validation of circular focal amplifications using Circle-Seq.** To further evaluate the accuracy of focal amplifications prediction from the AmpliconArchitect prediction, we chose ten pairs of GCAs from our cohort to perform ecDNA sequencing with Circle-seq[42] (see "Methods" section, Supplementary Fig. 5a). We performed ecDNA peak calling from Circle-seq using adjacent normal tissue as the control[43]. Among ten pairs of these selected GCA patients for Circle-Seq, four of them were circular focal amplifications positive by WGS prediction (Fig. 1b), and ecDNA amplicons (ranging from 491 to 39,020) were identified in all of them using Circle-Seq (Supplementary Fig. 5b). Then, we checked the overlapping ecDNA segments from Circle-seq and predicated ecDNA amplicons (circular focal amplifications) from the WGS in the four pairs of GCAs. We found that most ecDNA amplicons identified in the WGS appeared in the Circle-seq peak, where 100% WGS ecDNA in three GCAs, and 75% WGS ecDNA in one GCA were confirmed by Circle-seq (Fig. 2a). Since *CCNE1* was the most dominant detected focal amplifications across the cohort, we determined the detailed structure of *CCNE1* in Circle-seq (Supplementary Fig. 5c). We found that there was a clear enrichment of *CCNE1* in two GCAs from both Circle-seq and WGS, and that both had a similar tendency for amplification (Supplementary Fig. 5c). However, there was no *CCNE* amplification in the normal samples, in either WGS or Circle-seq, indicating that our circular focal amplifications detection, identified with AmpliconArchitect prediction from the WGS data, is reliable. The AA computational tool not only predicted the focal amplifications, but also provided the structure of the focal amplifications. Upon closer inspection comparing the fine structure of ecDNA amplification between the WGS and Circle-seq, we found that the fine structure was not always the same (Fig. 2b). The *FGFR2* ecDNA amplicon exhibited highly amplified

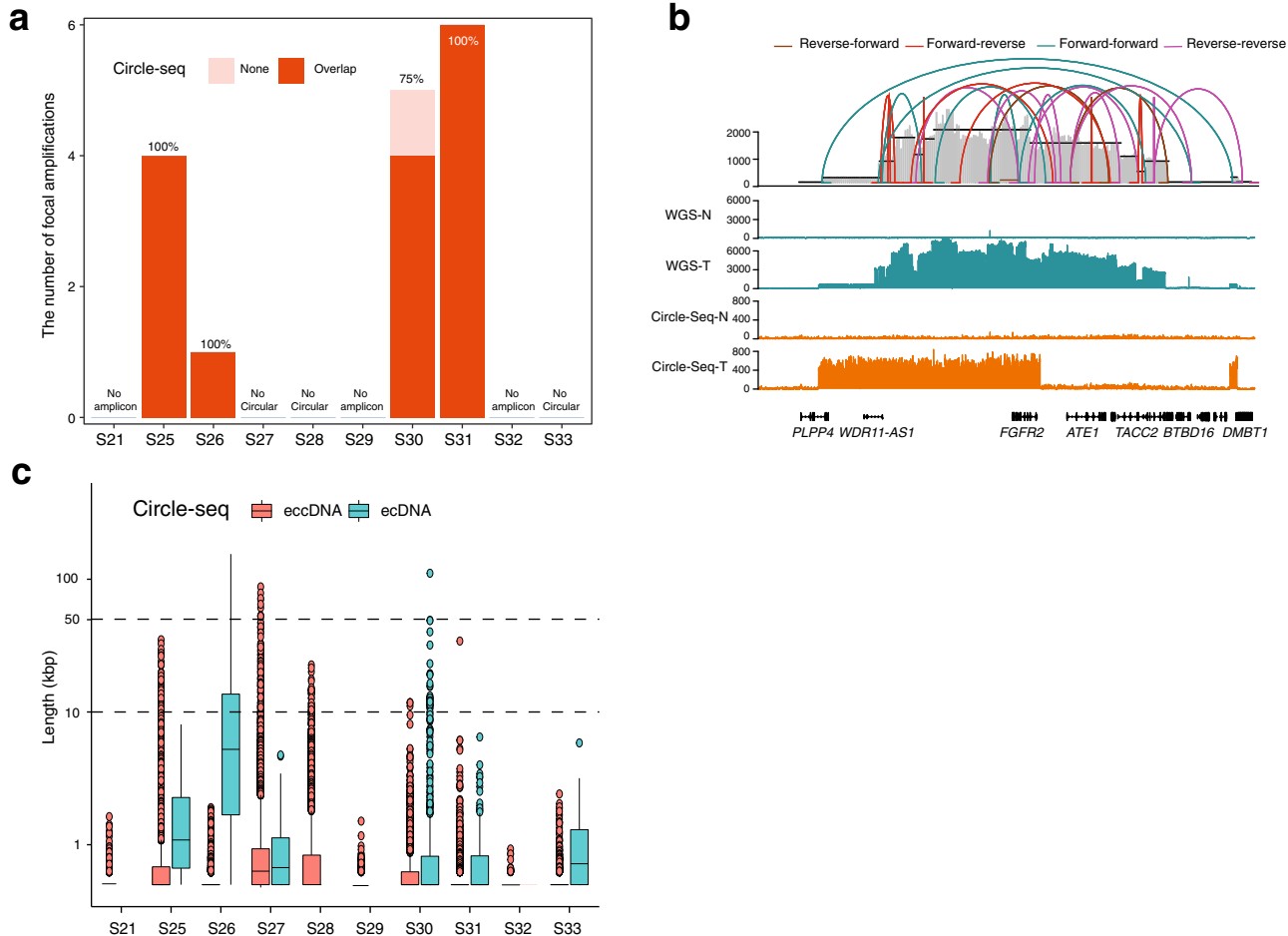

**Fig. 2 Validation of the ecDNA amplicons using Circle-seq. a** Summary of ecDNA overlapping lists from the prediction of AmpliconArchitect (AA) and identification using Circle-seq. The y-axis is the ecDNA amplicon number from WGS prediction. Overlap: the ecDNA amplicons were identified using both AA software from WGS and Circle-Seq. None: the ecDNA amplicons were only identified using AA software but not using Circle-Seq. **b** The genome browser track at the *FGFR2* gene locus from whole-genome sequencing (WGS) and Circle-seq. The connection lines on the top represent the potential structure combination in ecDNA amplicons predicted by AA software. N normal tissue, T tumor tissue. **c** Circular DNA elements identified from Circle-Seq were separated into ecDNA (copy number ≥ 7) (The number of ecDNA, $n = 0$, $n = 41$, $n = 42$, $n = 216$, $n = 0$, $n = 0$, $n = 714$, $n = 261$, $n = 0$, $n = 222$ for S21, S25, S26, S27, S28, S29, S30, S31, S32 and S33, respectively) and extrachromosomal circular DNA (eccDNA) (copy number < 7) (The number of eccDNA, $n = 1757$, $n = 7264$, $n = 4813$, $n = 38,804$, $n = 3610$, $n = 572$, $n = 2177$, $n = 3420$, $n = 481$, $n = 1686$ for S21, S25, S26, S27, S28, S29, S30, S31, S32 and S33, respectively). The box plots show the minima (bottom dot), the maxima (top dot), the median (middle line) and the first and third quartiles (boxes), whereas the whiskers show 1.5× the interquartile range IQR above and below the box. Source data are provided as a Source Data file for Fig. 2a–c.

segments with fluctuations in WGS prediction but not in the Circle-seq detection (Fig. 2b). The difference in the fine structure from WGS and Circle-seq likely reflects the technical bias of the ecDNA amplicon prediction from the WGS and library preparation from the Circle-seq. Furthermore, we separated the list of circular DNA elements from Circle-seq data into ecDNA (copy number ≥ 7) and extrachromosomal circular DNA (eccDNA) (copy number < 7) following previous report[44] (Fig. 2c). We found that all circular DNA elements from Circle-Seq in the four of six cases (S21, S28, S29, and S32), where WGS did not predict focal amplifications, are only from extrachromosomal circular DNA (eccDNA) (shorter than 50 kbp) (Fig. 2c). However, circular DNA elements from Circle-Seq in the other two of six cases (S27 and S33), where WGS did not predict focal amplifications either, contain both ecDNA and eccDNA (Fig. 2c).

**Focal amplifications in GCA are associated with chromothripsis.** Even though focal amplifications are widely detected in different types of cancer, the sources of focal amplifications remain unknown. It has been reported that chromothripsis

contributes to cancer progression and drives juxtaposition of rearranged segments of DNA from distinct chromosomal loci in cancer[3,45,46], and that some ecDNA amplicons and focal amplifications are generated during chromothripsis process[2]. Next, we aimed to understand the relationship between chromothripsis and focal amplifications in our GCA cohort. We used the ShatterSeek package[47] to identify chromothripsis events across the 36 GCA patients (Supplementary Fig. 6a). Strikingly, we found that chromothripsis occurred in 34 GCA patients across our cohort (Supplementary Fig. 6b). We also divided the chromothripsis events into fine categories with the parameters of high confidence (HC) and low confidence (LC) (see "Methods" section). This revealed that HC chromothripsis occurred in 61.1% of GCAs across the cohort, and LC chromothripsis occurred in 88.9% of all GCA samples. We found that the frequency of chromothripsis in GCA patients was quite diverse across the cohort, where the range of chromothripsis was from 0 to 4 for HCs and 0 to 14 for LCs (Supplementary Fig. 6c). The location of the chromothripsis events in the genome was also quite heterogeneous across the cohort (Fig. 3a). When we aligned chromothripsis events and

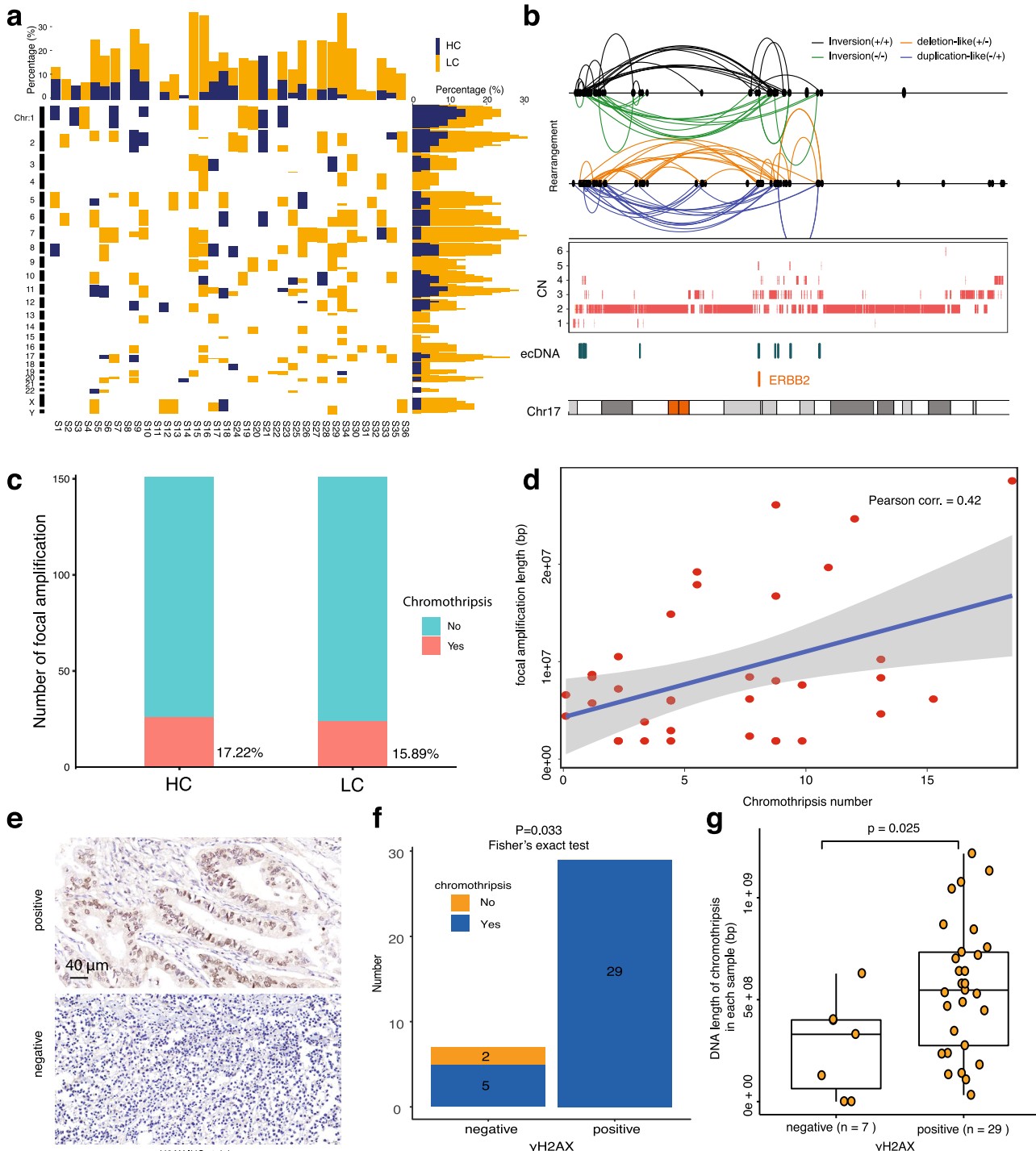

**Fig. 3 Focal amplifications and chromothripsis in GCA patients. a** Summary of chromothripsis events across the whole genome in our GCA cohort. HC high confidence chromothripsis, LC low confidence chromothripsis. **b** *ERBB2* focal amplifications in the event of chromothripsis from one GCA patient. The different connection lines on the top represent the potential different formats of chromothripsis events at the *ERBB2* gene. CN copy number. **c** Summary of overlapping frequency between focal amplifications and chromothripsis in the GCA cohort. HC high confidence chromothripsis, LC low confidence chromothripsis. **d** The correlation between total length of focal amplifications and the frequency of chromothripsis in GCA patients, where each dot represents one sample. **e** Representative images of γH2AX immunohistochemistry (IHC) staining in our GCA cohort. **f** Presence and absence of chromothripsis in γH2AX-positive and γH2AX-negative groups of GCA patients. The numbers on the bars are patient numbers. Two-sided Fisher exact test are performed. **g** Comparisons of the total length of chromothripsis in γH2AX-positive and γH2AX-negative GCA patients, where each dot represents one patient, and the length of chromothripsis is the total length of all chromosomes in each sample. The *p*-value was calculated using the two-sided Wilcoxon signed-rank test. The box plots show the minima (bottom dot), the maxima (top dot), the median (middle line) and the first and third quartiles (boxes), whereas the whiskers show 1.5× the interquartile range IQR above and below the box. Source data are provided as a Source Data file for Fig. 3a–d, f, g.

focal amplifications on the genome browser, we observed a clear overlap between focal amplifications and chromothripsis at some of the oncogene loci, including the *ERBB2* and *MYC* genes (Fig. 3b and Supplementary Fig. 7). To further explore the relationship between chromothripsis and focal amplifications, we quantified the number of focal amplifications that overlapped with chromothripsis (Fig. 3c). The results showed that 17.22% of focal amplifications occurred in HC chromothripsis, and 15.89% occurred in LC chromothripsis. Taken together, these results indicate that 33.11% of focal amplifications might be caused by chromothripsis (Fig. 3c). To further determine the relationship between focal amplifications and chromothripsis, we calculated the correlation between the number of chromothripsis events and the total length of all focal amplifications (Fig. 3d). The results clearly demonstrated a positive correlation between focal amplifications and chromothripsis events (Pearson's correlation = 0.42). Our results indicate the focal amplifications in GCAs are more likely to occur due to chromothripsis, and that such events could contribute to GCA progression if the chromothripsis event occurs at the oncogene site.

Comprehensive analysis of chromothripsis using large-scale samples of human cancers from TCGA showed that the frequency of chromothripsis is greater than 50% in several cancer types[48]. However, the frequency of chromothripsis in our GCA cohort was 94% (Fig. 3a), which is extremely high. Previous reports have shown that chromothripsis is associated with genomic instability and DNA damage[49–53]. Thus, we investigated potential risk factors contributing to such a high frequency of chromothripsis in our GCA cohort by analyzing genome stability and DNA damage. First, we performed microsatellite instability (MSI) detection by immunohistochemistry (IHC) staining of four proteins (MLH1, MSH2, MSH6, and PMS2)[54,55]. We found that only 9 of 36 samples were MSI-high samples (Supplementary Fig. 8a, b and Supplementary Data 4), and 27 patients were MSI-low. The two chromothripsis-negative samples were all in the MSI-low group (Supplementary Fig. 8b), and there was no correlation between MSI grade and chromothripsis events (Supplementary Fig. 8b, *p* = 1, Fisher's exact test). Thus, we concluded that the high frequency of chromothripsis is not likely due to the high proportion of MSI-high samples in our cohort. Second, we calculated chromosomal instability (CIN) for all 36 samples in accordance with a previous report[56] and divided GCA patients into four groups based on the genome integrity index (from low to high: 0–0.2, 0.2–0.4, 0.4–0.6, 0.6–0.8) (see "Methods" section). We found only two samples in our GCA patients in the high-grade CIN group (Supplementary Fig. 8c and Supplementary Data 4). The two chromothripsis-negative samples were in the low-grade CIN group (Supplementary Fig. 8c), and there was no correlation between CIN grade and chromothripsis events (Supplementary Fig. 8c, *p* = 0.381, Fisher's exact test). Thus, we concluded that the high frequency of chromothripsis is not likely due to the high proportion of high-grade CIN in our cohort. Third, we performed IHC staining of γH2AX protein, a crucial biomarker for the detection of DNA double strand breaks[57], in our GCA cohort. We found that 80.55% (29/36) of GCA patients were γH2AX protein positive (Fig. 3e, f and Supplementary Data 4). The two chromothripsis-negative samples were both γH2AX protein negative (Fig. 3f), and there was a significant correlation between the presence of γH2AX and chromothripsis events (Fig. 3f, *p* = 0.033, Fisher's exact test). We also found that the total length of chromothripsis in γH2AX protein-positive patients was significantly longer than that in γH2AX protein-negative patients (Fig. 3g, *p* = 0.025). Thus, we suspect that the high frequency of chromothripsis is most likely due to the high degree of DNA damage that has accumulated in GCA patients. All GCA patients in our study were

from the high incidence area for GCA in Henan Province, northern China[18], where the intake of nitrosamine-rich foods, such as pickled vegetables, has been well recognized as one of the key risk factors for GCA[58]. Accumulating evidence has demonstrated that nitrosamine is a very important factor for DNA alkylation, synthesis disorder, high instability and even DNA double strand breaks[59–64]. Thus, we suspected that nitrosamine exposure in our GCA cohort may accumulate DNA damage, potentially inducing a high frequency of chromothripsis. As ecDNA amplicons in our GCA cohort are more likely to occur due to chromothripsis, as stated above, and it was also proposed that chromothripsis is a primary mechanism that accelerates genomic DNA rearrangement and amplification into ecDNA by a recent study[3], our data suggest that local dietary habits from the geographic region in our cohort may contribute to focal amplifications occurrence in GCA patients.

**The presence of oncogene focal amplifications does not increase the mutation frequency in GCA.** Oncogene amplification is a key factor contributing to human cancer[65]. A high frequency of oncogene mutations has also been reported in GCA[29,34]. Since both oncogene amplification (Fig. 1d) and oncogene mutations (Supplementary Fig. 1b) were observed in our GCA cohort, we investigated whether there was a high frequency of oncogene mutations in the region of oncogene focal amplifications. We calculated numbers of SNVs in the whole genome as well as in only focal amplifications present regions (Supplementary Fig. 9a), and found mutation frequency in the focal amplifications regions occur at a similar level as in the whole genome from most patients, except for two GCA samples (Supplementary Fig. 9a). Statistical analysis showed that there was no significant difference in mutation frequency between focal amplifications regions and the whole genome in our GCA cohort (Supplementary Fig. 9b, *p* = 0.18). We also compared the numbers of SNVs in regions of individual oncogene focal amplifications regions (same oncogene observed in two or more patients) between present and absent oncogene focal amplifications patients (Supplementary Fig. 9c), and found that there were significantly more SNVs in the focal amplifications present group only with respect to the BIRC3 gene (Supplementary Fig. 9c, *p* = 0.031) but not at other oncogenes (Supplementary Fig. 9c). Thus, we concluded that there may be no relationship between oncogene mutations and the presence of oncogene focal amplifications in GCA patients.

**The presence of oncogene focal amplifications has the diverse correlations with the prognosis of GCA.** It was reported that the presence of ecDNA is associated with oncogene amplification and poor outcomes across multiple cancers[7]. Thus, we investigated the relationship between oncogene amplification, the presence of focal amplifications and patient prognosis in our GCA cohort. We first explored the relationship between oncogene amplification and GCA patient prognosis by focusing on the top 11 high frequency of oncogenes and TSGs ecDNA amplicons. We found that most of the top 11 high frequency oncogene amplifications across the cohort with a copy number (CN) greater than 5 came from focal amplifications (Supplementary Fig. 10). We compared the gene copy numbers and patient survival time by splitting the gene amplification into different groups (High, low, and normal) (Supplementary Fig. 10). As expected, the survival time in some GCA patients after surgery was shorter in those with a high copy number of certain oncogenes, including *EGFR*, *MYC*, and *BIRC3* (Supplementary Fig. 10). Surprisingly, we found that patients with a low CN amplification of *CCNE1* and *ERBB2* survived for a shorter period compared to those with a normal gene CN

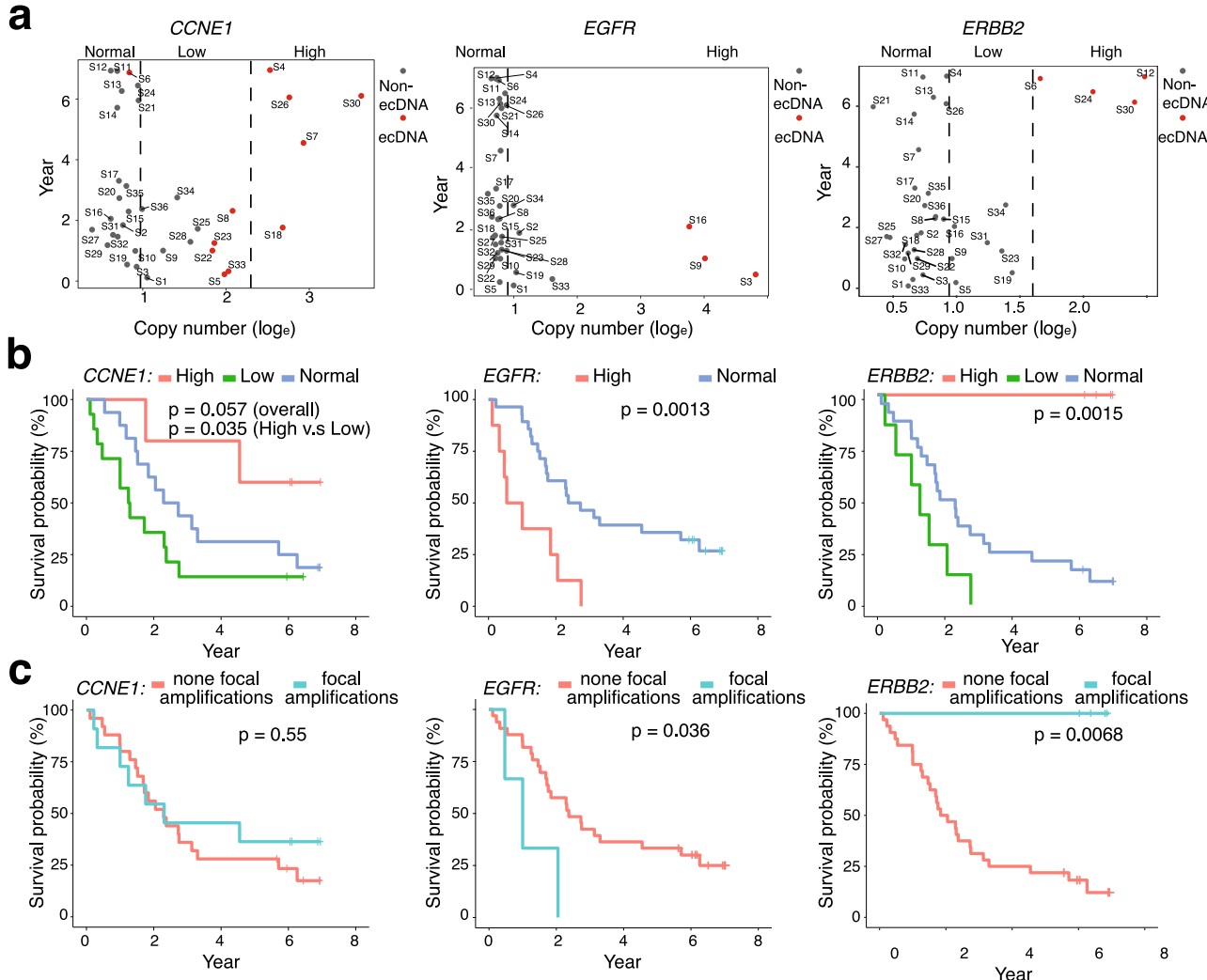

**Fig. 4 Oncogene amplification, focal amplifications presence and prognosis of GCA patients. a** The relationship between gene copy number and survival time for *CCNE1*, *EGFR*, and *ERBB2* genes in the GCA cohort, where copy number of *CCNE1* and ERBB2 genes were divided into three groups, high, low, and normal, and copy number of the *EGFR* gene was divided into two groups, high and normal. High high copy number of gene amplification, Low low copy number of gene amplification, Normal no gene amplification. **b** Survival analysis of different groups with three oncogene amplifications (*CCNE1*, *EGFR*, and *ERBB2*) in the cohort. The definition of high, low, and normal is the same as in **a**, and the *p*-value was calculated using the two-sided Log rank test. **c** Survival time of present and absent focal amplifications of three oncogenes in ecDNA (*CCNE1*, *EGFR*, and *ERBB2*) in the cohort. The *p*-value was calculated using the two-sided Log rank test. Source data are provided as a Source Data file for Fig. 4a–c.

(Supplementary Fig. 10), and patients survived even longer with a high CN of *CCNE1* and *ERBB2* amplification (Supplementary Fig. 10). To further investigate our observation, we performed a correlation study between different ranges of CN amplification and survival time from the *CCNE1*, *ERBB2*, and *EGFR* genes (Fig. 4a). The results indicated that the short survival time was due to the high range of oncogene amplification in *EGFR*. However, for *ERBB2* and part of the sample of *CCNE1*, the tendency was completely opposite. Specifically, we found that four samples with a high CN of *CCNE1*, caused by focal amplifications, exhibited an average survival time of 5.08 years, and all samples with a high CN of *ERBB2* had an average survival time of 6.59 years (Fig. 4a).

Furthermore, we focused on investigating the relationship between prognosis and CN of three oncogenes: *CCNE1*, *ERBB2*, and *EGFR*. *EGFR* followed the tendency that those with high-range oncogene amplification had a decreased survival time than those with low-range amplification (*p* = 0.0013) (Fig. 4b). The

relationship between *EGFR* copy number and patient survival time reflects oncogene function in tumorigenesis from GCAs. For both *ERBB2* and *CCNE1*, we found that patients with low range amplification had the worst prognosis compared to those with normal and high range amplification (Fig. 4b). To our surprise, patients with high range amplification from *CCNE1* and *ERBB2* had the best prognosis compared to those with low and middle range amplification (Fig. 4b). To further confirm the relationship between oncogene amplification and patient survival, we performed the WES sequencing on another independent GCA cohort with 39 GCA patients together with our 36 GCA patient cohorts (Supplementary Fig. 11a and Supplementary Data 5). First, the copy numbers of ERBB2 from WGS in the 36 patients were very similar to the copy numbers detected in the WES data (Supplementary Fig. 11b), which indicates that the WES data could be used to validate our WGS observation of ERBB2 gene amplification. Next, we focused on the WES data for 75 GCA patients, and we observed a similar tendency, namely, that the

high-range *ERBB2* amplification was correlated with increased survival time (Supplementary Fig. 11c and Supplementary Data 6). Taken together, we concluded that our observation is independent of the specific GCA cohort. Negative correlations between oncogene amplifications and patient prognosis have previously been reported in many independent studies, including large group studies in the TCGA[7]. We found a similar tendency for some oncogenes in GCA, such as *EGFR*. The negative correlation between oncogene amplifications and patients' prognoses is true for the low range amplification from *ERBB2* and *CCNE1* (Fig. 4b); however, the correlation becomes positive when these two genes undergo high range of amplifications (Fig. 4b).

Next, we investigated the relationship between the presence of oncogene focal amplifications and patient prognosis by dividing patients into focal amplifications present and absent groups (Fig. 4c), and we found diverse correlations of present oncogene focal amplifications and patient survival. In brief, we found no significant difference in prognosis for the absence and presence of *CCNE1* focal amplifications (Fig. 4c, $p = 0.55$); the presence of EGFR focal amplifications had a negative correlation with patient prognosis (Fig. 4c, $p = 0.036$); and the presence of ERBB2 focal amplifications had a positive correlation with patient prognosis (Fig. 4c, $p = 0.0068$). To understand whether our observation was due to clinicopathological factors from GCA patients, we first investigated the relationship between clinicopathological phenotypes and prognosis in GCA (see "Methods" section, Supplementary Fig. 12 and Supplementary Data 4). We found that UICC tumor stage was the only clinicopathological factor correlated with GCA survival (Supplementary Fig. 12i). Next, we performed survival analysis using clinicopathological variables of patients together with the presence of focal amplifications (*ERBB2, EGFR,* and *CCNE1*) by dividing patients into those with and without focal amplifications (Supplementary Fig. 12). We found that the presence of *ERBB2* focal amplifications may be relevant to the UICC tumor stage but not to other clinicopathological variables (Supplementary Fig. 12). However, the presence of *EGFR* and *CCNE1* focal amplifications was not relevant to any clinicopathological variables (Supplementary Fig. 12). Since both UICC tumor stage (Supplementary Fig. 12i) and the presence of focal amplifications (Fig. 4c) are contributing factors to patient survival, we assumed that there might be some connection between the presence of the *ERBB2* focal amplifications and GCA stage. However, our sample size was too small (36 cases) to obtain further conclusions. It will be very interesting to perform further studies with larger sample sizes of patients to obtain additional conclusions in the future.

The positive correlation between the presence of *ERBB2* focal amplifications in GCA and patient prognosis is paradoxical to large-scale TCGA studies in many cancer types[7], where the presence of ecDNA amplicons was shown to be associated with poor outcomes. Since it was reported that there is a paradoxical relationship between chromosomal instability and survival outcomes in cancer[56], we examined whether the positive correlation between the presence of *ERBB2* focal amplifications and patient prognosis is due to chromosomal instability (CIN) in our GCA cohort. The survival analysis from the four groups of CIN (see "Methods" section, Supplementary Fig. 8c) shows that GCA patients with stable chromosomes survived longer than patients with unstable chromosomes in our cohort (Supplementary Fig. 13a). However, we did not find that *ERBB2* focal amplifications present in samples were only enriched in specific CIN groups (Supplementary Fig. 13b), and we did not observe a significant difference in CIN values between focal amplifications present samples and focal amplifications absent samples (Supplementary Fig. 13c, $p = 0.33$). Thus, we concluded that the

paradoxical relationship between the presence of *ERBB2* focal amplifications in GCA patients and survival outcome is independent of CIN. A recent study showed chromatin structure of ecDNA is highly accessible[66], we assumed that the *ERBB2* gene may be highly expressed in ecDNA present GCA patients. It was also reported the amplification of *ERBB2* gene was followed by *ERBB2* gene overexpression in the same GCA tissue[27,67–69]. At the same time, we observed a positive correlation between *ERBB2* gene expression and ERBB2 protein expression in GCA patients ($n = 44$) (Supplementary Fig. 14a, $R = 0.79$, Supplementary Data 7). Thus, we hypothesized that protein levels of ERBB2 were also high in *ERBB2* focal amplifications present patients, and that a high level of ERBB2 protein would be positively associated with GCA prognosis. To test our hypothesis, we performed immunohistochemistry of the ERBB2 protein from 1668 GCA patients (with 0-year to 7-year survival time after surgery) (see "Methods" section, Supplementary Fig. 14b and Supplementary Data 8). We found there is significant difference of surviving probabilities in ERBB2 positive and negative patients ($p$ value = 0.024 with Fleming Harrington function ($p = 1$, $q = 1$)) (Supplementary Fig. 14c), where the survival probability of ERBB2 positive patients was lower than that of ERBB2 negative patients when their surviving time is under 2 years; however, the tendency became opposite when their surviving time is longer than 2 years. It was reported ERBB2 protein expression and gene amplification correlate with better survival in esophageal adenocarcinoma[70]. Our observation that the survival probability of ERBB2 positive patients (when their surviving time is longer than 2 years) was longer than that of ERBB2 negative patients in our GCA cohort, probably also reflects the similarity between esophageal adenocarcinoma features and GCA. Since we assumed that the protein level of ERBB2 is high in *ERBB2* focal amplification positive patients, our observations indicate that the *ERBB2* focal amplifications probably represent a good prognostic marker in GCA patients with surviving time longer than 2 years.

## Discussion

In this work, we identified focal amplifications and ecDNA amplicons in GCA patients using WGS data, and validated these ecDNA amplicons using Circle-seq. We found that these focal amplifications are present in most GCA patients, and have exhibit heterogeneity in different GCA patients. Additionally, we found that several oncogenes are in the format of focal amplifications in GCA patients and that different oncogenes could coamplify in the same focal amplifications. Interestingly, we found oncogene focal amplifications were associated with a high frequency of chromothripsis in our GCA cohort, and such a high frequency of chromothripsis in our cohort is likely due to high degree of DNA damage induced by nitrosamine exposure from a local diet[59–64]. We propose that local dietary habits from the geographic region may have contributed to focal amplifications occurrence in our GCA cohort. It was reported that focal amplifications in nearly half of the samples across a variety of cancer types can be explained by ecDNA formation[12,17] and ecDNA is a major mechanism of drug resistance in several tumor types[3], thus, it will be valuable to follow clinical annotation on previous exposure therapy together with ecDNA detection in large-scale samples of GCA patients to design therapy strategies for GCA patients in the future.

Strikingly, we found that the correlation between the present oncogene focal amplifications and patient prognosis was different depending on gene in GCA patients, where *ERBB2* focal amplifications positively correlated with prognosis, *EGFR* focal amplifications negatively correlated with prognosis and *CCNE1* focal amplifications did not correlate with prognosis. The correlation

between presence of focal amplifications and prognosis in GCA reported in this study is different from a previous report indicating that oncogene ecDNA amplicons correlate with poor prognosis in other cancers from TCGA[7], and our observation likely reflects the heterogeneous nature of cancers. These diverse correlations of oncogene focal amplifications and prognosis may aid in designing better personal therapy strategies for GCA patients in the future. Large-scale ERBB2 immunohistochemistry results showed survival probability of ERBB2 positive patients was lower than that of ERBB2 positive patients when their surviving time is under 2 years, however, the tendency became opposite when their surviving time is longer than 2 years. Since we assumed that the protein level of ERBB2 is high in ERBB2 focal amplifications positive patients, our observations indicate that the ERBB2 focal amplifications may represent a good prognostic marker in GCA patients with surviving time longer than 2 years.

## Methods

All clinical samples were collected following the ethic permit from the local hospitals (An Yang cancer hospital, China and the First Affiliated Hospital, Zhengzhou University, China) located at high-incidence areas of GCA in the Taihang Mountains of north central China. The ethical research committees at An Yang cancer hospital, China and the First Affiliated Hospital, Zhengzhou University, China approved the study. All patients were informed in our study with a consent document signed, and the effect that consent to publish clinical information potentially identifying individuals was obtained.

**GCA samples collection and follow-up visiting of patients.** All patients in our study were not received radiotherapy or chemotherapy before the surgery. 1668 GCA patients for ERBB2 immunohistochemistry (IHC) staining are from the Esophageal Cancer database (from years of 1973–2020) which established and maintained by Henan Key Laboratory for Esophageal Cancer Research of the First Affiliated Hospital, Zhengzhou University, China[19,20,27,60]. In our Esophageal cancer database, Clinical GCA tumors and matched normal tissues are both preserved with snap freezing in liquid nitrogen and archived in formalin-fixed paraffin-embedded (FFPE) tissue block for each GCA patient. In the studies of whole genome sequencing (WGS), whole exome sequencing (WES), RNA-Seq, and protein expression measurement with mass spectrometry, snap freezing samples were used. In the study of IHC staining, FFPE samples were applied. The diagnosis of GCA patients were always identified by two well-trained pathologists in the pathology department of the local hospital, where the hematoxylin and eosin (HE) staining was used to quantify the content of tumor cell in tissue section and only GCA samples with more than 80% tumor cells are used for our study. The matched normal tissue samples were selected from the adjacent epithelial tissue which is 5–10 cm away from the edge of tumor. Both of 36 pairs of GCA tumor and matched adjacent normal tissue for whole-genome sequencing (WGS) and 75 pairs of GCA tumor and matched adjacent normal tissue for whole-exon sequencing (WES) are scanned and confirmed with two well-trained pathologists in the same procedure. The complete clinicopathological information of all patients was recorded and included in our study. All patients are included in regular follow-up visiting plan with following frequency: once every 3 months during the first year, once each 6 months during the second year, and once per year after the third year. The definition of overall survival time for dead patients is a period from diagnosis to death, and the definition of overall survival time for alive patients is a period from diagnosis to last follow-up visit (Jan 2021).

**WGS library preparation and sequencing.** WGS sequencing libraries were prepare following the previous report with slight modifications[71].

In brief, genomic DNA was extracted from snap freezing GCA tumor and matched normal tissue with DNeasy Blood & Tissue Kit (69504, QIAGEN) following manufacturer instruction. DNA concentration was measured by Qubit DNA Assay Kit in Qubit 2.0 Flurometer (Invitrogen). A total amount of 0.4 μg DNA per sample was fragmented to an average size of ~350 bp with hydrodynamic shearing system (Covaris, Massachusetts, USA) and subjected to DNA library preparation with Illumina TruSeq DNA sample preparation kit (15026486, Illumina). Sequencing was carried out on Illumina NovaSeq 6000 with 150 bp paired end mode according to the manufacturer instruction.

**WES library preparation and sequencing.** WES sequencing libraries were prepare following the previous report with slight modifications[72].

In brief, genomic DNA was extracted from snap freezing GCA tumor or matched normal tissue using DNeasy Blood & Tissue Kit (69504, QIAGEN) according to the manufacturer's instruction. DNA degradation and contamination were monitored on 1% agarose gels. DNA concentration was measured by Qubit

DNA Assay Kit in Qubit 2.0 Flurometer (Invitrogen). A total amount of 0.6 μg genomic DNA per sample was fragmented to an average size of 180–280 bp and subjected to DNA library preparation using Illumina TruSeq DNA sample preparation kit. The Agilent SureSelect Human All ExonV5 Kit (5190-6209, Agilent Technologies) was used for exome capture according to the manufacturer's instruction. In brief, DNA libraries were hybridized with liquid phase with biotin labeled probes from the Agilent SureSelect Human All ExonV5 Kit, then magnetic streptavidin beads were used to capture the exons of genes. Captured DNA fragments were enriched in a PCR reaction with index barcodes for sequencing. Final libraries were purified using AMPure XP beads (A63880, Beckman Coulter) and quantified using the Agilent high sensitivity DNA kit (5067-4626, Agilent Technologies). WES libraries were sequenced on Illumina Novaseq 6000 (Illumina) with 150 bp paired end mode according to the manufacturer instruction.

**Circle-Seq library preparation and sequencing.** EcDNA sequencing Service was provided by CloudSeq Biotech Inc. (Shanghai, China) by following the published procedures with slight modification[73]. Circle-Seq was performed on ten pairs of snap freezing GCA tumors and matched normal tissues. In brief, 6 mg of snap freezing GCA tumors or matched normal tissues tissue were suspended in L1 solution (A&A Biotechnology, 010-50) and supplemented with 15 μl proteinase K (ThermoFisher, E00491) before incubation overnight at 50 °C with agitation. After lysis, samples were alkaline treated, followed by precipitation of proteins and separation of chromosomal DNA from circular DNA through an ion exchange membrane column (Plasmid Mini AX; A&A Biotechnology, 010-50). Column-purified DNA was treated with FastDigest MssI (ER1341, Thermo Scientific,) to remove mitochondrial circular DNA and incubated at 37 °C for 16 h. Remaining linear DNA was removed by exonuclease (E3101K, Plasmid-Safe ATP-dependent DNase, Epicenter,) at 37 °C in a heating block and enzyme reaction was carried out continuously for 1 week, adding additional ATP and DNase every 24 h (30 units per day) according to the manufacturer's protocol (E3101K, Plasmid-Safe ATP-dependent DNase, Epicenter,). ecDNA-enriched samples were used as template for phi29 polymerase amplification reactions (150043, REPLI-g Midi Kit) amplifying ecDNA at 30 °C for 2 days (46–48 h). Phi29-amplified DNA was sheared by sonication (Bioruptor), and the fragmented DNA was subjected to library preparation with NEBNext® Ultra II DNA Library Prep Kit for Illumina (E7645S, New England Biolabs). Sequencing was carried out on Illumina NovaSeq 6000 with 150 bp paired end mode.

**ERBB2 RNA expression measurement and ERBB2 protein expression measurement in GCA patients.** ERBB2 RNA expression measurement and ERBB2 protein expression measurement in 44 GCA patients from our Esophageal Cancer database (from years of 1973–2020), where ERBB2 RNA expression (Normalized value with RPKM (Reads Per Kilobase Million)) was extracted from RNA-seq data, and ERBB2 protein expression was extracted from mass spectrometry. For same GCA patient, both library for RNA-seq and library for mass spectrometry (MS) are prepared. The procedures of libraries preparation are briefly described as below. For RNA-seq library preparation: First, 100 mg of each snap freezing GCA tumor tissue was used for total RNA isolation with TRIzol® Reagent (15596026, Thermo Fisher Scientific). RNA purity was checked using the NanoPhotometer® spectrophotometer (IMPLEN). RNA concentration was measured using Qubit® RNA Assay Kit in Qubit® 2.0 Flurometer (Life Technologies). RNA integrity was assessed using the Bioanalyzer 2100 system (Agilent Technologies). Then, two RNA-seq libraries were prepared for each GCA patients with technical replicates. Fifty nanogram total RNA was used as input for each RNA library preparation. The RNA-Seq libraries were prepared with NEBNext® UltraTM RNA Library Prep Kit for Illumina (E7530L, NEB) by following manufacturer's instruction. RNA-seq libraries were purified with AMPure XP beads (A63880, Beckman Coulter) to select 150–200 bp cDNA fragments. Sequencing library was quantified on the Bioanalyzer 2100 system (Agilent Technologies). The libraries were sequenced on an Illumina Novaseq 6000 platform with 150 bp paired-end reads. The RNA-seq sequencing libraries were aligned to the genome using STAR[74] with default parameter to reference genome (hg19). After the alignment, the ERBB2 RNA expression are extracted, and normalized with RPKM. The final expression data for individual patient used to compare with protein expression is the average value of two technical replicates. For mass spectrometry library preparation: First, 10 mg of snap freezing GCA tumor tissues were grinded with liquid nitrogen into powder and then transferred to a 5 ml centrifuge tube. After that, four volumes of lysis buffer (1% Triton X-100, 1% protease inhibitor cocktail, and 1% phosphatase inhibitor) was added to the cell powder, followed by sonication three times on ice using a high intensity ultrasonic processor (Scientz). The remaining debris was removed by centrifugation at $12,000 \times g$ at 4 °C for 10 min. After centrifugation, the supernatant was collected and the protein concentration was measured with Piece™ BCA protein kit (23227, Thermo Fisher Scientific) according to the manufacturer's instruction. Then, the 100 μg of protein from each sample was taken for protein digestion, and the volume was adjusted to the same with lysate. The sample was slowly added to the final concentration of 20% v/v trichloroacetic acid (TCA) to precipitate protein, then vortexed to mix and incubated for 2 h at 4 °C. The precipitated protein was collected by centrifugation at $4500 \times g$ for 5 min at 4 °C. The precipitated protein was washed with pre-cooled acetone for three times to remove traces of TCA and finally acetone was removed by drying in a fume cupboard. The

protein sample was then added 100 mM Triethylammonium bicarbonate (TEAB) and ultrasonically dispersed. Trypsin was added at 1:50 trypsin-to-protein mass ratio for the first digestion overnight. The sample was reduced with 5 mM dithiothreitol for 30 min at 56 °C and alkylated with 11 mM iodoacetamide for 15 min at room temperature in darkness. Next, 50 μg of tryptic peptides were firstly dissolved in 0.5 M TEAB. Each channel of peptide was labeled with their respective TMT reagent, and incubated for 2 h at room temperature. Five microliters of each sample were pooled, desalted, and analyzed by MS to check labeling efficiency. After labeling efficiency check, samples were quenched by adding 5% hydro-xylamine. The pooled samples were then desalted with Strata X C18 SPE column (Phenomenex) and dried by vacuum centrifugation. Then, the dried tryptic peptides were dissolved in solvent A (0.1% formic acid, 2% acetonitrile/in water), directly loaded onto a home-made reversed-phase analytical column (25 cm length, 100 μm i.d.). Peptides were separated with a gradient from 8% to 10% solvent B (0.1% formic acid in 90% acetonitrile) over 2 min, 10–23% solvent B over 38 min, 23–33% in 14 min and climbing to 80% in 3 min then holding at 80% for the last 3 min, all at a constant flowrate of 450 nl/min on an EASY-nLC 1200 UPLC system (Thermo Fisher Scientific). The separated peptides were analyzed in Q ExactiveTM HF-X (Thermo Fisher Scientific) with a nano-electrospray ion source. The electrospray voltage applied was 2.2 kV. The full MS scan resolution was set to 120,000 for a scan range of 400–1500 $m/z$. Up to 20 most abundant precursors were then selected for further MS/MS analyses with 30 s dynamic exclusion. The HCD fragmentation was performed at a normalized collision energy (NCE) of 28%. The fragments were detected in the Orbitrap at a resolution of 45,000. Fixed first mass was set as 100 $m/z$. Automatic gain control (AGC) target was set to 5E4, with an intensity threshold of 5.8E4 and a maximum injection time of 86 ms. The resulting MS/MS data were processed using MaxQuant search engine (v.1.6.10.43). Tandem mass spectra were searched against the human SwissProt database (20366 entries) concatenated with reverse decoy database. Trypsin/P was specified as cleavage enzyme allowing up to two missing cleavages. The mass tolerance for precursor ions was set as 10 ppm in First search and 5 ppm in Main search, and the mass tolerance for fragment ions was set as 0.02 Da. Carbamidomethyl on Cys was specified as fixed modification. Acetylation on protein N-terminal, oxidation on Met and deamidation (NQ) were specified as variable modifications. TMT-11plex quantification was performed. FDR was adjusted to <1% and minimum score for peptides was set >40. The ERBB2 protein expression level for each patient was extracted from protein lists of MS result.

**Data analysis of WGS data, WES data, copy number alteration (CNA), and focal amplifications calling**. All detailed scripts were deposited in following link: https://github.com/chenlab2019/ecDNA-on-GCA; https://zenodo.org/record/5544035#.YV3PJi0Rp0K[75]. The WGS data of 36 samples were aligned to the reference genome (hg19) using BWA-MEM v.0.7.17[76] with the default parameter and were sorted by SAMtools v.1.9[77]. PCR duplicates were removed from aligned BAM files by Sambamba v.0.7.0[78]. By taking matched normal samples as background, tumor-specific CNAs were called by copyCat package (https://github.com/chrisamiller/copyCat) which is loosely based on readDepth[79]. During the process of CNA calling, bam-window tools (https://github.com/genome-vendor/bam-window) was used to count reads in 10 Kbp window size. AA was applied to filter CNAs with copy number greater than 4× and size greater than 100 Kbp. The adjacent CNAs were merged into a single interval. These intervals were fed into AA software[9] as seeds to detect focal amplifications[30]. The genomic annotation of focal amplifications amplicons was performed with intersection between regions of focal amplifications and genomic annotation of reference genome (hg19) with bedtools[80]. In brief, regions of the focal amplifications were extracted from the output of AA software. The intersection between genomic annotation of reference genome (hg19) and focal amplifications regions was performed with bedtools first[80], then the length of overlapping regions between genomic elements from reference genome and focal amplifications regions was extracted. Genomic elements were annotated to focal amplifications amplicons if there was 1 bp or longer overlapping. The occupancy of coding regions and exons regions in focal amplifications amplicons were calculated with following formulas:

$$\text{Occupancy of coding regions in ecDNA (\%)} = \frac{\text{Total length of coding regions in all focal amplifications amplicons}}{\text{Total length of all focal amplifications amplicons}} \times 100, \quad (1)$$

$$\text{Occupancy of exon regions in ecDNA (\%)} = \frac{\text{Total length of exon regions in all focal amplifications amplicons}}{\text{Total length of all focal amplifications amplicons}} \times 100. \quad (2)$$

Focal amplifications amplicons were further classified into different categories (linear, complex, circular, breakage-fusion-bridge (BFB), and invalid) with AA software (https://github.com/jluebeck/AmpliconClassifier) by following the previous report[7]. Circle-finder[35–37] was used to confirm the circular structure of focal amplifications amplicons by following the instruction, where circular junction points were detected with sequencing reads orientation. The length of overlapping

region between circular focal amplifications predicted from AA and circular focal amplifications detected with Circle-finder was calculated with bedtools. When the length of overlapping region is longer than 1 bp, circular focal amplifications amplicons from AA were labeled as overlapping with results of Circle-finder.

For WES data analysis from 75 pairs of GCA tumor samples and matched adjacent normal tissues: sequencing reads containing adaptors and low-quality reads were removed and aligned to human reference genome (hg19) using BWA-MEM v.0.7.17[76] with the default parameter and sorted by SAMtools v.1.9[77]. All non-primary alignments were filtered by SAMtools. PCR duplicates were marked using Picard. CNAs from tumor was called by using matched adjacent normal tissues by CNVkit[81]. The numbers of CNAs on ERBB2 gene from each GCA patient are extracted for further analysis.

**Data mining of Circle-seq**. All reads were aligned to human genome hg19 using BWA-MEM v.0.7.17[76] with default parameters. PCR duplicates were removed from the BAM file with Sambamba v.0.7.0[76]. By taking normal samples as background, peak calling on tumor samples was performed using variable-width windows of Homer v.4.11 with command *findPeaks tumor -i normal -style histone -fdr 0.001* (http://homer.ucsd.edu/)[82]. The tumor-specific enriched peaks were considered as the fragments of circular DNA. The circular DNA elements detected from Circle-Seq were separated into ecDNA (copynumber > 7) and extrachromosomal circular DNA (eccDNA) (copy number <7) following previous report[83]. Overlaps between enriched peaks from Circle-Seq and focal amplificationsfrom AA were calculated, and circular focal amplifications from AA is labeled as validated when the overlapping regions is 1 bp or longer than 1 bp. For the visualization of the peak of Circle-seq, BAM file was converted into bigwig file using deeptools bamCoverage with normalization of counts per million (CPM)[84].

**Detection of chromothripsis events**. All detailed scripts were deposited in following link: https://github.com/chenlab2019/ecDNA-on-GCA. Chromothripsis events from 36 pairs of GCA tumor samples were detected with ShatterSeek software v.0.4 using copy number alterations (CNAs) and structural variants (SVs) following the previous report[47]. SVs were identified on tumor samples using the Delly[85] and novoBreak[86] software by taking matched adjacent normal tissues as control, and final list of SVs are merged lists from Delly and NovoBreak. CNAs from WGS were calculated with copyCat package[87]. All SVs and CNVs from tumor samples are used to identify chromothripsis events with ShatterSeek, where SVs and CNVs from matched adjacent normal tissues are treated as background. Events were considered as high confidence (termed HC) when there were at least seven oscillating CN segments, and considered as low confidence (termed LC) when there were 4–6 oscillating CN segments[11]. The chromothripsis events were labeled as within regions of focal amplificationswhen there is 1 bp or longer intersection between segments from chromothripsis and regions of focal amplifications.

**Single-nucleotide variant (SNV) analysis**. All detailed scripts were deposited in following link: https://github.com/chenlab2019/ecDNA-on-GCA. All SNVs from WGS were called by GATK v.4.1.7 software[88] with Mutect2 parameter and filtered by "GATK FilterMutectCalls". The mutation profiles were visualized by R/Bio-conductor package "maftools"[89]. The number of SNVs within region of focal amplifications and whole genome region were counted respectively for each sample. The average number of SNVs per million nucleotides from regions of focal amplifications and whole genome were calculated with following equations:

$$\text{SNVs of ecDNA} = \frac{\text{The number of SNVs in ecDNA amplicons}}{\text{The total length of ecDNA amplicons}} \times 1\text{ million}, \quad (3)$$

$$\text{SNVs of whole genome} = \frac{\text{The number of SNVs within whole genome}}{\text{The total length of whole genome}} \times 1\text{ million}. \quad (4)$$

Numbers of SNVs within individual oncogene focal amplifications from groups of absent and present this gene focal amplifications were also compared: first high frequency of oncogene focal amplifications (appeared at least in two patients) in 36 patients are selected, then the number of SNVs within each selected oncogene from individual patient was calculated and numbers of SNVs between groups of present and absent this oncogene focal amplifications were compared. Six hundred and sixty-six cancer driver genes were extracted from previous reports[31–33].

**Oncogene annotations and oncogene focal amplifications analysis**. All detailed scripts were deposited in following link: https://github.com/chenlab2019/ecDNA-on-GCA. In brief, oncogene annotation in focal amplifications was performed with AA software[12] and AmpliconClassifier[83] (https://github.com/jluebeck/AmpliconClassifier), where graph and cycles files generated by AA were taken by AmpliconClassifier. A table indicating which genes are present on the focal amplifications regions was generated from AmpliconClassifier, and the list of canonical oncogenes[31–33] from the gene table were chosen as the oncogenes list in the focal amplifications. The full oncogenes or truncated oncogenes presenting on the focal amplification regions was checked by intersection between genomic coordinates of oncogenes and genomic interval of focal amplification with bedtools. In our GCA cohort, it showed 85% of oncogenes (98 of 115 oncogenes) listed by

AmpliconClassifier were fully carried in the focal amplifications, and only 17 of 115 oncogenes are with truncated—5′ or 3′ end in the focal amplification (Supplementary Fig. 3a). The list of oncogenes focal amplifications was extracted from the report of AmpliconArchitect following AmpliconArchitect workflow[9]. The copy number of each oncogene from 36 GCA samples was extracted from the report of copyCat. Oncogenes are labeled as oncogene co-amplification if two or more than two oncogenes and/or tumor suppressor genes are located in the same focal amplification.

*Calculation of chromosomal instability (CIN)*. All detailed scripts were deposited in following link: https://github.com/chenlab2019/ecDNA-on-GCA. The CIN was calculated following the previous report[56], and groups of CIN is defined with by number of genome integrity index (GII). GII was defined as the fraction of the genome that was altered based on the common regions of alteration. CIN of GCA patients was divided into four groups based on GII (0–0.2, 0.2–0.4, 0.4–0.6, 0.6–0.8), and 36 GCA patients were assigned into different groups of CIN.

**Prognoses and statistical analysis**. All computational codes aand scripts are deposited in following link: https://github.com/chenlab2019/ecDNA-on-GCA. R package "survival" with Kaplan–Meier method was used[90] to calculate and compare patient prognosis between different groups of GCA patients. The statistic methods used in prognosis analysis with clinicopathological factor are as follows: Fisher's exact test for sex, family history cigarette smoking, alcohol consuming and tumor stage, and Wilcoxon signed-rank test for age. All analyses were performed on R v.3.6.2, Python v.2.7.16 and Python 3.7.4. The visualization of survival curve was conducted by ggplot2[91], karyoploteR[92], pheatmap R packages and Circos[93], IGV software[94]. The Rényi family test was adopted when two KM curves acrossed[95,96]. The Rényi test was conducted with R package "survMisc"[97].

**Immunohistochemistry (IHC) staining of ERBB2 protein**. IHC was performed by following the previous report[98] with slightly modifications. In brief, 5 µm thick formalin fixed paraffin-embedded GCA tissue sections were first deparaffined with xylene 15 min for three times, then were dehydrated through 100% alcohol, 85% alcohol and 75% alcohol for 5 min each, followed by distilled water rinsing for 5 min. The epitope retrieval is performed in the microware by putting the tissue into citrate buffer (pH 6.0). After the epitope retrieval, the tissue section is rinsed in Phosphate-Buffered Saline buffer (PBS, pH 7.4). After blocked with 3% bovine serum albumin (BSA) 30 min at room temperature, the tissues were incubated with ERBB2 antibody (1:100 dilution, SAB5700151, Sigma-Aldrich) overnight at 4 °C. In the next day, the washing is performed with PBS buffer for three times, 15 min each. The secondary antibody (1:1000 dilution, Horseradish Peroxidase, HRP marked, PV-9000, ZSGB-BIO) was incubated for 50 min at room temperature. After the secondary antibody incubation, the washing is performed with PBS buffer three times on shaker, 15 min each. The tissue is stained with the Harris Hematoxylin for 3 min. At last, the tissue section was mounted and imaged. Sections with no signal in any cell were defined as negative groups; sections with five or more cells with ERBB2 positive signal were defined as positive groups.

*IHC staining of γH2AX*. The staining protocol is same as ERBB2 staining. The primary antibody of γH2AX (SAB5700329, Sigma-Aldrich) was with 1:200 dilution.

The staining of γH2AX was categories into positive and negative groups with following parameters: Section with no γH2AX signal in any cell was defined as γH2AX negative groups; section with five or more cells with γH2AX positive signal was defined as γH2AX positive groups.

*MSI detection with IHC staining*. IHC staining of four mismatch repair (NMR) proteins: MLH1 (1:100 dilution, PA5-32497, Thermo Fisher Scientific), MSH2 (1:500 dilution, MA5-15740, Thermo Fisher Scientific), MSH6 (1:100 dilution, MA5-32040, Thermo Fisher Scientific) and PMS2 (1:150 dilution, MA5-26269, Thermo Fisher Scientific), were performed on 5 µm thick FFPE tumor sections from 36 GCA patient with same protocol as stated as above in ERBB2 IHC staining. The patient was labeled as microsatellite instability (MSI-high) if one of NMR proteins was negative stained, otherwise the patient is labeled as MSI-low.

**Reporting summary**. Further information on research design is available in the Nature Research Reporting Summary linked to this article.

## Data availability

The raw data of WGS data, WES data and Circle-Seq data generated in this study have been deposited in the China National Center for Bioinformation under accession code HRA000814. Source data are provided with this paper.

## Code availability

All detailed scripts used in this study were deposited in following link: https://github.com/chenlab2019/ecDNA-on-GCA; https://zenodo.org/record/5544035#.YV3PJi0Rp0K[75].

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

## Acknowledgements

This work is supported by the National Key R&D Program of China (2016YFC0901403 to L.D.W.), the National natural science foundation of China (81872032, U1804262 to L.D.W.), Henan Medical Science and technology research project (LHGJ20190001 to X.K.Z.), the Swedish Research Council (VR-2016-06794, VR-2017-02074 to X.C.), Beijer Foundation (to X.C.), Jeassons Foundation (to X.C.), Petrus och Augusta Hedlunds Stiftelse (to X.C.), Göran Gustafsson's prize for younger researchers (to X.C.), Vleugel Foundation (to X.C.), and Uppsala University (to X.C.).

## Author contributions

L.D.W. and X.C. conceived and designed the study; X.K.Z., X.S., L.L.L., R.H.X., W.L.H., P.P.W. and F.Y.Z. contributed to the collection of the patient materials and clinical information; X.K.Z., X.S., M.M.Y., J.F.H. and K.Z. prepared the WGS and ecDNA sequencing of GCA; P.X. performed all the sequencing data mining; L.Z., Y.D., L.L.L., X.N.H., C.L.M. and J.J.J. were responsible for the protein expressions of ERBB2, γH2AX, and MSI staining in the GCA and analysis of the relationship with the GCA survival; M.Z., X.K.Z., P.X., L.D.W. and X.C. wrote the manuscript together with input from all authors; and L.D.W. and X.C. supervised all aspects of this work.

## Funding

## Competing interests

The authors declare no competing interests.
