## [Peer review file · Nature Communications]

REVIEWER COMMENTS

Reviewer #1 (Remarks to the Author): Expert in computational genomics and ecDNA

Zhao et al performed whole-genome sequencing analysis of a tumor type with high frequency in specific areas of China, gastric cardia adenocarcinoma. In particular, they focused on the detected of circular amplifications, also extrachromosomal amplifications or ecDNA. EcDNAs were found at very high frequency, in all but one of the 36 patient tumors. Circle-seq was employed to validate WGS-based ecDNA calls, with high but not perfect accuracy. This is an interesting manuscript on a rapidly developing area of cancer research, namely extrachromosomal oncogene amplifications. The following comments aim to further improve the manuscript.

Major comments

1. Details on whole-genome sequencing, such as coverage and QC, would benefit the manuscript.
2. Were all amplicons reported by the computational method classified as circular and were any linear or complex linear structural rearrangements identified as well? Figure 1b suggests that each tumor contains numerous circular DNA amplifications which is surprising. Can the authors include representative example figures in the supplement to provide visual validation of circular DNA calls?
3. WGS detected circular DNA in 35/36 but circleseq detected circular DNA in 7 out of ten. This suggests that too many WGS-derived amplicons are labeled as circular. Additional in silico verification of AA calls is needed, to ensure their circular nature.
4. A wide range in circular amplicon sizes are reported, up to 42.8Mbs. The upper bound seems very large for an amplicon. Can the possibility that these very large structures can be deconvoluted into multiple structurally independent circular amplicons be excluded?
5. 2452 sites are reported as protein coding regions; what does this mean? That in total, 2452 genes were detected an any of the circular amplicons? Figure 1C requires more detail to be interpretable; in fact, the written legends for all figures are very brief and expanding them would clarify the figures. Figure 2A is another example: what does "overlap" mean, and what does "none" indicate?
6. Features of oncogenes amplified ecDNAs are compared to TCGA gastric and esophageal cancer, concluding that GCA is more similar to gastric cancer. This statement requires testing for significance.
7. It is not clear what Figure 2A is showing. The number of ecDNAs counted across the ten samples analyzed by Circleseq ranges from 491 to 39020 , whereas the 2A y-axis is in the dozen range.
8. All but one samples in this n=36 cohort contain one or more ecDNAs, and there are many different survival analyses performed. This complicates the interpretation in terms of whether circular amplification confers worse prognosis, as has been suggested.
9. The frequency of ecDNA reported here is higher than any of the cancer types analyzed by Kim et al, Nat Genetics, 2020. Gastric cardia adenocarcinoma is reported to be at high prevalence in specific parts of China, in particular the Henan province. Can the authors speculate on gastric cardia adenocarcinoma risk factors which could drive genomic instability and/or the extrachromosomal DNA amplification frequency reported here?

Minor comments:

1. Please clarify what is meant by the following statements: "However, the frequency of these ecDNA amplicons is higher in the coding region (6.28%) compared to the whole genome (3.48%) (Supplementary Fig. 1g). Furthermore, the proportion of ecDNA amplicons detected in the exons is higher (14.5%) compared to the whole genome (9.2%) (Supplementary Fig. 1h)."
2. Overexpression of CCNE1 is mentioned, but there is no mention of RNA expression data for the cohort.

Reviewer #2 (Remarks to the Author): Expert in cancer genomics

In this manuscript, Zhao et al. investigated the presence and the role of extrachromosomal circular DNA (ecDNA) in a cohort of 36 patients with gastric cardia adenocarcinoma. They used whole-genome sequencing of tumor and adjacent normal DNA and the AmpliconArchitech algorithm to assess the

presence of ecDNA. They explored the role of ecDNA by annotating amplicons and focused their attention on oncogenes. They used ShatterSeek to explore the role of chromothripsis in the etiology of ecDNA and claimed that a third of scDNA might be explained by chromothripsis. Finally, they investigate the impact of CCNE1, EGFR and ERBB2 copy number on survival and used a larger cohort to confirm that HER2 overexpression is associated with survival. The manuscript presents several major flaws.

Major:

Generally, the manuscript is not well written and sometimes not intelligible. The figure legends are incomplete. The study design (patients' selection, follow-up time) is lacking and the methodology (WGS, ecDNA prediction) is not rigorous enough to draw any valid conclusions. Importantly, the study is not reproducible as the original data and the code to regenerate the results are not available.

The method section could benefit from a more rigorous and detailed description. Crucial details on the samples used for this study is lacking. For example, was the DNA extracted from FFPE or frozen samples? This is important for subsequent analysis as DNA extracted from FFPE is often degraded by the fixation method. Were the tumor and normal samples assessed by a trained pathologist to assess tumor content and the absence of tumor cells in the normal adjacent tissues?

The quality of the sequencing is hard to assess. Apart from the sequencing depth (for which the method to compute is not described) present in figure S1a, no other metrics are present. For example, it would have been interesting to see if the classical driver mutations (TP53, CDH1, PIK3CA) present in GCA were also found in the present study.

Basic clinicopathological variables of the patients included in this study are lacking (age at diagnosis, sex, grade, stage, histology, molecular classification, EBV status). Among other important omissions, the specific library used for the WGS is not detailed and the WES from the independent 75 GCA patients is not described. It is unclear how exactly the authors have annotated the genomic elements present in the predicted ecDNA.

The timepoints used for overall survival are not defined. The statistic to assess the impact of ecDNA on survival is incorrect. In general, survival analysis is meaningless without taking into consideration clinicopathological variables of patients (what are the characteristics of patients with and without ecDNA?)

In Figure 1e and the text, the authors describe the presence of ecDNA measuring 40Mb. To put this in context the whole chromosome 21 measures 47Mb. It was previously reported that the size of ecDNA in human cancer was between 1 and 3Mb (<https://www.nature.com/articles/s41568-019-0128-6>). Could the authors comment on this big difference?

The reported frequency of chromothripsis event in the present study is 35/36 (97%) which is surprisingly high compared to what was reported using the same method (50%, <https://www.nature.com/articles/s41588-019-0576-7>). Could the authors comment? Did the cohort include some MSI-high/CIN tumors?

It is unclear if the cohort of 1668 patients used to assess the impact of HER2 amplification on survival was done in the context of this study. Does the IHC and pathological assessment were done to corroborate the results or does the data was preexisting? Could the authors provide more details on how they assessed the presence/absence of HER2 amplification? There is a discrepancy between the number of patients referred in the text (n=1668) and the legend of figure 4d (n=1598). Furthermore, the p-value indicated in the text and figure 4d (p=0.027) is not the same as the one in figure S7 (p=0.16). Supplementary table: only survival time is present (the unit is absent, is it years, months?). The event status is not present whereas in figure 4d it seems that some patients are censored. Could

the author address those issues?

Correlative analysis of clinical features of patients and specific molecular subtypes (EBV, MSI, CIN, GS) could strengthen the study. Extrachromosomal DNA is a major mechanism of drug resistance in several tumor types (<https://www.nature.com/articles/s41586-020-03064-z>). No clinical annotation on previous exposure therapy is currently present but it would be interesting to explore this in more detail.

Minor:

The introduction and discussion are not titled.

Figure 3d. The number of chromothripsis events goes from 0 to 20. In the text, the upper range is 14 (line 166)

Line 33. What "diverse correlation with patients' prognosis" means here

Line 44. What other "human disease" please clarify

Line 51-52. Please explain the different features associated with GCA from Chinese pop? Citation from 1987, are there any more recent epidemiologic studies?

Line 62. CNV is related to germline variations. Please use CNAs when referring to somatic.

Line 68-71. The size of ecDNAs are reported to be from 100kb to 22.6Mb (line 68) and then from 62kb to 42.8Mb (line 71). It is very unclear if the authors describe the sum of all ecDNA per sample or the real size.

Suppl Figure 1f. the Barplot is not needed.

Line 89. "Overexpression" more specific to protein and should not be used to describe the amplification of a gene.

Line 96. CDK12 is not an oncogene but a TSG.

Line 100-103. This sentence is not intelligible.

Line 119. What "computational tool" please detailed.

Line 235. Table number is X.

Line 399. Please provide the exact reference of the antibody used (ID, manufacturer).

Figure 4c. There is no "low" category for the EGFR survival analysis.

Reviewer #3 (Remarks to the Author): Expert in gastric cancer genomics and clinical research

In this study, Zhao and colleagues report a survey of extrachromosomal DNA (ecDNA) in gastric cardia adenocarcinoma (GCA). Analyzing WGS data of 36 GCAs, they find that different patients have distinct levels of ecDNA amplicons, with varying sizes. Several of the amplicons involve known oncogenes (eg CCNE1, HER2/ERBB2), and the overall amplicon profiles are similar to gastric cancer compared to esophageal cancer. The ecDNA amplicons are enriched in certain genomic features, such as exons and protein-coding genes. Certain oncogenes (EGFR, CDK6) are found in the same ecDNA amplicon. A number of the WGS-inferred ecDNA amplicons were confirmed by circle-Seq. There is an association between specific ecDNA amplicons and regions of chromothripsis. They also report for CCNE1 and HER2 amplicons, that patients with very high levels have paradoxically improved survival.

1) The finding that multiple oncogenes are found in the same ecDNA amplicon is quite interesting, as it provides an obvious mechanism for oncogene co-amplification. How often does this type of pairwise interaction occur?

2) Can the authors investigate if there is any relationship between gene mutations and/or somatic copy number changes in the 'regular' nuclear genome and the presence of ecDNAs? For example, do patients with HER2 ecDNA amplicons also exhibit HER2 regular copy number amplifications?

3) Have the authors considered that the relationship between survival and HER2 ecDNA may relate to

overall patterns of genomic instability, as reported in earlier studies such as Birkbak et al., 2011 Cancer Research (PMID: 21270108).

Overview

We thank the reviewers for their enthusiastic assessment of our manuscript and thoughtful comments that have improved the manuscript. Following suggestions from three reviewers, we added the following main points to our revised manuscript:

- 1,** The manuscript underwent English language editing by a professional language editing firm.
- 2,** Our computational code and scripts in the manuscript and all of our raw data are deposited in the publicly accessible platform.
- 3,** The detailed QCs of WGS data were further performed from two angles: **i)** we performed a detailed QC analysis of sequencing depth, coverage, sequencing quality, and mapping rate and included these QC data in the revised **Supplementary Table 1**; and **ii)** we calculated the gene mutation frequency from WGS in our cohort and compared the list of top mutated genes with classical driver mutation genes reported in previous GCA research^{2,3,4} and included this result in the revised **Supplementary Figure 1b**.
- 4,** Detailed categorization of ecDNA amplicons¹, including linear, complex, circular, BFB and invalid amplicons, was performed and have been added to our revised manuscript (revised **Supplementary Table 2**, revised **Figure 1b**).
- 5,** We verified the circular features of circular ecDNA identified using AmpliconArchitect (AA) software with another in silico method, Circle-finder (a method to identify circular DNA from paired-end high-throughput sequencing data, https://github.com/pk7zuva/Circle_finder)⁵⁻⁷, by checking the sequencing read orientation and junction points of circular ecDNA. We found that 89.94-100% of circular ecDNA amplicons identified from AA contain the junction reads detected using Circle-finder (revised **Supplementary Figure 1f-h**).
- 6,** The detailed characterization of ecDNA amplicon size was added (revised **Supplementary Figure 2b**) in our revised manuscript.
- 7,** We investigated oncogene ecDNA co-amplifications from the list of oncogene and tumour suppressor gene ecDNA amplicons (**Lines 182-207**, revised manuscript). We found **i)** co-amplification of oncogene ecDNA (two or more oncogenes and/or tumour suppressor genes in the same ecDNA amplicon) occurred in 50% of patients (18 of 36 patients) (revised **Figure 1e**); **ii)** the frequency of oncogene ecDNA co-amplification varied from 50% to 100% in all oncogene amplifications in different patients (revised **Figure 1e**, revised **Supplementary Figure 4a**); and **iii)** some pairs of oncogene ecDNA co-amplification were observed in more than one patient (revised **Supplementary Table 3**), where pairs of *ERBB2* and *CDK12*, *RARA* and *SMARCE1*, *CBLC* and *BCL3* occurred in 3 patients; pairs of *EGFR* and *IRF4*, *PPARG* and *RAF1*; and pairs of *CDK12*, *ERBB2* and *RARA* occurred in 2 patients.
- 8,** We investigated the potential reasons for the high frequency of chromothripsis in our GCA cohort from three angles:
 - i),** We performed microsatellite instability (MSI) detection using immunohistochemistry (IHC) staining of four proteins (MLH1, MSH2, MSH6 and PMS2)^{8,9}. We found that only 9 of 36 samples were MSI-high samples (revised **Supplementary Figure 8a, 8b**, revised **Supplementary Table 4**), while 27 patients were MSI-low. The two chromothripsis-negative samples were both in the MSI-low group (revised **Supplementary Figure 8b**), and there was no correlation between MSI grade and chromothripsis events (revised **Supplementary Figure 8b**, $p = 1$, Fisher's exact test). Thus, we concluded that the high frequency of chromothripsis is not likely due to the high proportion of MSI-high samples in our cohort. We have included this part of the study in the revised manuscript (**Lines 278-285**, revised manuscript).

ii), We followed the same method in Birkbak et al., 2011 Cancer Research¹⁰ (PMID: 21270108) to calculate the overall pattern of chromosomal instability (CIN) for all 36 samples and divided our GCA patients into four groups based on the genome integrity index (from low to high: 0 to 0.2, 0.2 to 0.4, 0.4 to 0.6, 0.6-0.8). Only 2 samples were in the high-grade CIN group (revised **Supplementary Figure 8c**, revised **Supplementary Table 4**). The two chromothripsis-negative samples were both in the low-grade CIN group (revised **Supplementary Figure 8c**), and there was no correlation between CIN grade and chromothripsis events (revised **Supplementary Figure 8c**, $p = 0.381$, Fisher's exact test). Thus, we concluded that the high frequency of chromothripsis is not likely due to the high proportion of high-grade CIN in our cohort. We have included this part of the study in the revised manuscript (**Lines 285-293**, revised manuscript).

iii), We performed immunohistochemistry (IHC) staining of γ H2AX protein, a crucial biomarker for the detection of DNA double strand breaks¹¹, in our GCA cohort. We found that 80.55% (29/36) of GCA patients were γ H2AX protein positive (revised **Fig. 3e, 3f**, **Supplementary Table 4**). The two chromothripsis-negative samples were both γ H2AX protein negative (revised **Fig. 3f**), and there was a significant correlation between the presence of γ H2AX and chromothripsis events (revised **Fig. 3f**, $p = 0.033$, Fisher's exact test). We also found that the total length of chromothripsis from the genome was significantly longer (revised **Fig. 3g**, $p = 0.025$, revised **Supplementary Table 4**) in γ H2AX protein-positive patients than in γ H2AX protein-negative patients. Thus, we think the high frequency of chromothripsis in our GCA cohort is most likely due to the GCA patients in our study being from a high incidence area for GCA in Henan Province, northern China¹², where the intake of nitrosamine-rich foods, such as pickled vegetables, has been well recognized as one of the key risk factors for GCA¹³. Accumulating evidence has demonstrated that nitrosamine is a very important factor for DNA alkylation, synthesis disorder, high instability and even DNA double strand breaks¹⁴⁻¹⁹. Thus, we suspected that nitrosamine exposure in this population may accumulate DNA damage and further trigger the high frequency of chromothripsis. We have included this part of the study in the revised manuscript (**Lines 293-313**, revised manuscript).

9, We investigated the relationship between single-nucleotide variants (SNVs) and the presence of ecDNA amplicons. We quantified the number of SNVs within the regions of ecDNA amplicons and the whole genome and compared the number of SNVs from ecDNA regions and the whole genome. We found that the frequency of SNVs observed in ecDNA- regions was in a similar range as that in the whole genome (revised **Supplementary Figure 9a, 9b**, $P = 0.18$). We also compared the numbers of SNVs in regions of individual oncogene regions and tumour suppressor gene regions (same oncogene or tumour suppressor gene ecDNA observed in 2 or more patients) between the presence and absence of oncogene ecDNA groups (revised **Supplementary Figure 9c**) and found that there were significantly more SNVs in the ecDNA present group only with respect to the BIRC3 gene (revised **Supplementary Figure 9c**, $p = 0.031$) but not from other oncogenes. Our conclusion is that there is likely no relationship between gene mutations and the presence of oncogene ecDNA amplicons.

10, The detailed procedure of clinical sample collection has been added to the revised **Materials and Methods**, and the detailed clinicopathological information of the patients is included in the revised **Supplementary Table 4**. The clinicopathological information was taken into consideration for the survival analysis with present and absent ecDNA amplicons (revised **Supplementary Figure 12**). We added one more survival curve based on the presence and absence of three oncogene ecDNA amplicons (*ERBB2*, *EGFR* and *CCNE1*) to the main figure (revised **Figure 4c**). These curves clearly show that there was no difference in patient prognosis in the absence and presence of *CCNE1* ecDNA amplicons; the presence of *EGFR* ecDNA amplicons had a negative correlation with patient prognosis (revised **Figure 4c**, $p = 0.036$), and the presence of *ERBB2* ecDNA amplicons had a positive correlation with patient prognosis (revised **Figure 4c**, $p = 0.0068$). Our conclusion is that there are diverse correlations between the presence of ecDNA oncogene and prognosis in GCA.

11, We investigated the relationship of chromosomal instability (CIN), survival and the presence of *ERBB2* ecDNA amplicons (revised **Supplementary Figure 13**). We found that longer survival in the presence of *ERBB2* ecDNA amplicons was independent of the CIN grade of patients.

We answer reviewer comments point-by-point below.

Reviewer #1 (Remarks to the Author): Expert in computational genomics and ecDNA

Zhao et al performed whole-genome sequencing analysis of a tumor type with high frequency in specific areas of China, gastric cardia adenocarcinoma. In particular, they focused on the detected of circular amplifications, also extrachromosomal amplifications or ecDNA. EcDNAs were found at very high frequency, in all but one of the 36 patient tumors. Circle-seq was employed to validate WGS-based ecDNA calls, with high but not perfect accuracy. This is an interesting manuscript on a rapidly developing area of cancer research, namely extrachromosomal oncogene amplifications.

Thank you for the positive feedback and thoughtful comments. We share the reviewer’s enthusiasm for the interesting observations of extrachromosomal oncogene amplifications in gastric cardia adenocarcinoma research.

The following comments aim to further improve the manuscript.

Major comments

1. Details on whole-genome sequencing, such as coverage and QC, would benefit the manuscript.

Thank you for your suggestion. In our first submission of the manuscript, we only showed the sequencing coverage of our whole-genome sequencing data with a bar graph in Supplementary Figure S1a. In our revised manuscript, the detailed QCs of WGS data were further performed from two angles: **i)** we performed a detailed QC of our sequencing data, including sequencing depth, coverage, sequencing quality, and mapping rate, and have included these QC data in the **revised Supplementary Table 1**; and **ii)** we calculated the gene mutation frequency from WGS and compared the top mutated genes list with classical driver mutated genes present in previous research on GCA² and have included this result (**Lines 92-95**, revised manuscript) in the revised **Supplementary Figure 1b**.

Supplementary Table 1

Sample ID	Reads after removing duplicates	>Q20 (Percentage)	>Q30 (Percentage)	Mapping_rate	Coverage
S1_Tumor	33838927950	0.95	0.89	99.93%	10.91
S2_Tumor	38874477450	0.95	0.89	99.90%	12.53
S3_Tumor	32234315250	0.95	0.89	99.91%	10.39
S4_Tumor	37106553150	0.95	0.90	99.93%	11.96
S5_Tumor	79017403950	0.95	0.88	99.93%	25.47
S6_Tumor	65857136100	0.95	0.88	99.70%	21.23
S7_Tumor	65666901300	0.95	0.89	99.76%	21.17
S8_Tumor	75468588300	0.94	0.87	96.13%	24.33
S9_Tumor	69980563050	0.95	0.88	99.38%	22.56
S10_Tumor	90095794350	0.95	0.89	99.65%	29.05
S11_Tumor	86715235350	0.96	0.90	99.62%	27.96
S12_Tumor	76455384750	0.96	0.91	99.59%	24.65

Supplementary Figure 1b

QC of whole-genome sequencing data from GCA patients. Part of **Supplementary Table 1**: Detailed QC of WGS data; **Supplementary Figure 1b**: Top mutated genes identified from 36 pairs of GCA cohorts with whole-genome sequencing data.

2. Were all amplicons reported by the computational method classified as circular and were any linear or complex linear structural rearrangements identified as well? Figure 1b suggests that each tumor contains numerous circular DNA amplifications which is surprising. Can the authors include representative example figures in the supplement to provide visual validation of circular DNA calls?

Thank you. This is a really good question. We apologize for the unclear description of the ecDNA concept in our previous version of the manuscript. In our manuscript, we followed the concept of ecDNA (extrachromosomal amplifications) from previous reports^{20,21} (PMID:31748743, 30674876) to describe our ecDNA amplicons in GCA patients, which contain both linear and circular DNA amplification. The summary of ecDNA amplicons from all patients in our previous Figure 1b includes both linear and circular ecDNA. To avoid misunderstanding, we further classified ecDNA amplicons into four categories (linear, complex, circular and BFB) using AA software²¹ and have included this detailed characterization in the revised **Figure 1b**, revised **Supplementary Figure 1c** and revised **Supplementary Table 2** of our revised manuscript. Figure 1e of the previous manuscript shows one case of circular ecDNA identified with AA software. It is good suggestion to include more examples of circular ecDNA. In our revised manuscript, we have included another two cases (revised **Supplementary Figure 1d** and **1e**).

Figure 1b

Supplementary Figure 1c

Supplementary Figure 1d

Supplementary Figure 1e

Classification of ecDNA amplicons (revised Figure 1b and revised Supplementary Figure 1c) and more examples of visual validation of circular DNA from AA software (revised Supplementary Figure 1d and revised Supplementary Figure 1e).
--

3. WGS detected circular DNA in 35/36 but circleseq detected circular DNA in 7 out of ten. This suggests that too many WGS-derived amplicons are labeled as circular. Additional in silico verification of AA calls is needed, to ensure their circular nature.

We apologize again for the misleading numbers and concepts in the first submission of the manuscript and the incorrect citation of the figure (Line 135, Figure 2a should be supplementary figure 3b). Here, we provide a detailed explanation, which is included in the revised manuscript. The positive rate of ecDNA amplicons with WGS prediction was approximately 77.78% (28/36) (Supplementary Figure 1b of the previous version), and circular ecDNAs were identified using Circle-seq in all 10 cases (Supplementary Figure 3b of the previous version).

In lines 132-133 of the previous version, we stated “Among these 10 pairs of GCAs, ecDNA amplicons were identified in seven of them (**Fig. 2a**)”. This sentence is inaccurate and incomplete, which also misleads the reviewer, and the citation of the figure is not accurate either. Our real intention was as follows: Among 10 pairs of these selected GCA patients for Circle-Seq, seven of them were ecDNA amplicon positive by WGS prediction (**Fig. 1b**), and ecDNA amplicons (ranging from 491 to 39020) were identified in all of them using Circle-Seq (**Supplementary Figure 5b**). To avoid misleading readers, we have updated the statement (**Lines 213-216**, revised manuscript) in our revised manuscript and performed language proofreading using a professional language firm.

It is a very good suggestion to use another in silico method to validate our finding of circular ecDNA from the WGS prediction. In the revised manuscript, we verified the circular features of circular ecDNA identified using AmpliconArchitect (AA) software with another in silico method, Circle-finder (a method to identify circular DNA from paired-end high-throughput sequencing data, https://github.com/pk7zuva/Circle_finder)⁵⁻⁷, by checking the sequencing read orientation and junction points of circular ecDNA. We found that 89.94-100% of circular ecDNA amplicons identified from AA contain the junction reads detected using Circle-finder (revised **Supplementary Figure 1f**). This result is included in our revised manuscript (**Lines 114 to 119**, revised manuscript). We have also included two examples of circular ecDNA amplicons detected by both Circle-finder and AA software (revised **Supplementary Figure 1g** and **Supplementary Figure 1h**).

Circle-finder was used to validate the circular features of circular ecDNA amplicons identified using AA software.

Supplementary Figure 1f: Summary of overlapping frequency in circular ecDNA amplicons from prediction of AmpliconArchitect (AA) and detection using Circle-finder. The empty bar represents no ecDNA or no circular ecDNA amplicons identified by AA; **Supplementary Figure 1g, Supplementary Figure 1h:** Example of circular ecDNA amplicon detected using both AA software and Circle-finder. Bottom panel: The location of forward and reverse sequencing reads (R1 = read 1, R2 = read 2) in the circular junction point are indicated on the genome browser.

4. A wide range in circular amplicon sizes are reported, up to 42.8Mbs. The upper bound seems very large for an amplicon.

We apologize for the unclear description in the previous version, which has been updated in the revised version. The size of the largest ecDNA amplicon was 22.6 Mb. The 42.8 Mb was the total amplicon length on chromosome 19. To avoid misunderstanding, we removed the summary of total ecDNA amplicon size of each chromosome (it is not very informative). In our revised manuscript, we characterized the size of ecDNA amplicons in the following format: **i)** size of ecDNA amplicons in each sample (revised **Supplementary Figure 2a**); **ii)** the size distribution of ecDNA amplicons from all samples (revised **Supplementary Figure 2b**), where 75% of ecDNA amplicons were smaller than 1 Mbp, 90% of ecDNA amplicons were in the range of 0-5 Mbp, 3% of ecDNA amplicons were larger than 10 Mbp, and only 1% of ecDNA amplicons were larger than 20 Mb; **iii)** the size of ecDNA amplicons in each chromosome (revised **Supplementary Figure 2d**).

Supplementary Figure 2a

Supplementary Figure 2b

Supplementary Figure 2d

Characterizing the size of ecDNA amplicons. **Supplementary Figure 2a:** Summary of ecDNA amplicon size in each GCA patient, where each dot represents one ecDNA amplicon; **Supplementary Figure 2b:** Size distribution of ecDNA amplicons in our GCA cohort; **Supplementary Figure 2d:** Summary of ecDNA amplicon size in each chromosome from the cohort, and each dot represents one ecDNA amplicon.

Can the possibility that these very large structures can be deconvoluted into multiple structurally independent circular amplicons be excluded?

That is a good question. We used AA software to deconvolute the possible combination of fine structures from long ecDNA amplicons (larger than 20 Mbp) and found that there were multiple potential combinations for some ecDNA amplicons (revised **Supplementary Figure 2c**). Thus, we could not exclude the possibility that a very large structure could be deconvoluted into multiple structurally independent circular amplicons. We have included this point in the revised manuscript (**Lines 126-128** of the revised manuscript).

Supplementary Figure 2c

Example of large ecDNA amplicons (> 20 Mbp) deconvoluted into multiple potential combinations of amplicons using AA software, where different connection lines on the top represent potential combinations within ecDNA amplicon regions (revised Supplementary Figure 2c).

5. 2452 sites are reported as protein coding regions; what does this mean? That in total, 2452 genes were detected in any of the circular amplicons? Figure 1C requires more detail to be interpretable; in fact, the written legends for all figures are very brief and expanding them would clarify the figures. Figure 2A is another example: what does “overlap” mean, and what does “none” indicate?

Thank you for pointing out our unclear description text and figure legends. We performed language proofreading using a professional English-proofreading firm. All of our figure legends have been updated with a detailed description and were confirmed by the English proofreading firm. For these two cases indicated by the reviewer: **i)**, 2452 genes were detected in all ecDNA amplicons; **ii)**, Overlap: the ecDNA amplicons were identified using both AA software and Circle-Seq; none: the ecDNA amplicons were only identified using AA software but not with Circle-Seq.

6. Features of oncogenes amplified ecDNAs are compared to TCGA gastric and esophageal cancer, concluding that GCA is more similar to gastric cancer. This statement requires testing for significance.

Thanks for your comments. The sentence, “However, the oncogene ecDNA ranking list is significantly different between oesophageal cancer and GCAs (**Supplementary Figure 2**), Line 109-110”, in our previous manuscript is not intelligible. We apologize for the nonprofessional language, which has been updated in the revised manuscript. What we intended to show in our previous manuscript is as follows: since the gastric cardia is located at the junction of gastric and oesophageal cancer, we compared the top ranking lists of ecDNA amplicons from these three cancer types, gastric cardia adenocarcinoma (our study), gastric cancer (TCGA) and oesophageal cancer (TCGA), and determined the differences and similarities in ecDNA amplicons in these three types of cancers. Our conclusion is that the ecDNA amplicon ranking list from gastric cancer is similar to that of gastric cardia adenocarcinoma but not to that of oesophageal cancer.

We adjusted our statement in the revised manuscript as follows:

Lines 175-177 in revised manuscript: However, the top ranking list of oncogene ecDNA amplicons was different between esophageal cancer and GCAs

Lines 178-179 in revised manuscript: Our results indicate that the top oncogene ecDNA amplicons from GCAs is more similar to those from gastric cancer.

7. It is not clear what Figure 2A is showing. The number of ecDNAs counted across the ten samples analyzed by Circleseq ranges from 491 to 39020, whereas the 2A y-axis is in the dozen range.

Thank you for your comments. Our unclear description of the figure legends misleads the reviewer again, which has now been updated. The y-axis in Figure 2a is the ecDNA amplicon number from WGS prediction, not from Circle-seq. The number of ecDNAs detected by Circle-Seq is shown in the revised **Supplementary Figure 5b**. In Figure 2a, we compared the overlapping number of ecDNA amplicons from WGS prediction and Circle-Seq. The Y-axis shows the number of ecDNA amplicons from WGS prediction, where green indicates that there are overlaps between the amplicons and circle-seq regions, and red indicates that there are no overlaps between them. We updated our figure legends and made this point clearly.

8. All but one samples in this n=36 cohort contain one or more ecDNAs, and there are many different survival analyses performed. This complicates the interpretation in terms of whether circular amplification confers worse prognosis, as has been suggested.

Thank you for your comments. We apologize again that our inaccurate description may have confused the reviewer, and our unprofessional organization of Figure 4 complicated the analysis.

Here, we intended to show the following in our first version of the manuscript: First, we investigated the relationship of oncogene amplification and GCA patient prognosis by focusing on the top 11 high frequency oncogene ecDNA amplicons and compared the oncogene copy numbers and patient survival by splitting the gene amplification into different copy number (CN) groups (High, Low, Normal) (original **Figure 4a**, revised **Supplementary Figure 10**). We found that patients survived longer with a high CN of *CCNE1* and *ERBB2* amplification, where 4 patients (from ecDNA amplicons) in our cohort with a high CN of *CCNE1* had an average survival time of 5.08 years, and all samples (n = 4, from ecDNA amplicons) with a high CN of *ERBB2* had an average survival time of 6.59 years (**Fig. 4b**); however, patients with a high CN of EGFR exhibited reduced survival. To make this point clearer and more precise, we made several changes in our revised manuscript:

- i) updated our language to avoid any misleading statements;
- ii) followed reviewer 2's suggestion to add clinicopathological information to our survival analysis (revised **Supplementary Figure 12**) to make our conclusion more solid;
- iii) moved previous Figure 4a to the revised **Supplementary Figure 10** and focused only on the prognostic analysis in revised Figure 4. In the revised Figure 4, **Figure 4a-b** shows the relationship between prognosis and oncogene (*ERBB2*, *EGFR* and *CCNE1*) amplification; **Figure 4c** presents the relationship between prognosis and the presence of oncogene ecDNA (*ERBB2*, *EGFR* and *CCNE1*); and **Figure 4d** shows the relationship between prognosis and ERBB2 IHC staining from a large population of GCA patients. The survival curve in revised **Figure 4c** clearly shows that there is no significant difference in prognosis in the absence or presence of *CCNE1* ecDNA (p = 0.55); the presence of *EGFR* ecDNA had a negative correlation with prognosis (revised **Figure 4c**, p = 0.036); and the presence of *ERBB2* ecDNA had a positive correlation with prognosis (revised **Figure 4c**, p = 0.0068). Thus, we concluded that there are diverse correlations between the presence of ecDNA oncogene amplicons and prognosis.

With these modifications, the relationship between the presence of oncogene ecDNA amplicons and prognosis is now clear.

Figure 4c

Survival time analysis based on the presence and absence of ecDNA amplicons of three oncogenes (*CCNE1*, *EGFR*, and *ERBB2*) in the cohort (revised Figure 4c). The p-value was calculated using the log rank test.

9. The frequency of ecDNA reported here is higher than any of the cancer types analyzed by Kim et al, Nat Genetics, 2020.

Thank you for your comments. Perhaps our unclear description misleads reviewers again, and we apologize for that. The frequency of ecDNA in our study is in a similar range to that reported in TCGA papers¹. The frequency of ecDNA amplification in our GCA cohort was 77.8% (28 samples from 36 patients), which is similar to that of oesophageal cancer (approximately 80%) but higher than that of gastric cancer (approximately 50%)¹ (Nat. Genetics paper, 2020). We have added this point to our revised manuscript (Lines 102-104, revised manuscript).

Frequency of circular amplification across tumour and nontumour tissues (Kim et al, Nat Genetics, 2020)¹

Gastric cardia adenocarcinoma is reported to be at high prevalence in specific parts of China, in particular the Henan province. Can the authors speculate on gastric cardia adenocarcinoma risk factors which could drive genomic instability and/or the extrachromosomal DNA amplification frequency reported here?

This is a great suggestion, which was suggested by the other two reviewers as well. In our previous version of the manuscript, we only investigated the relationship between genome chromothripsis and ecDNA amplicons, and our results indicate that ecDNA amplicons in GCAs are more likely to occur because of chromothripsis. However, we did not investigate the potential risk factors for chromothripsis.

In our revised manuscript, we investigated the potential reasons for the high frequency of chromothripsis in our GCA cohort from three angles:

i), We performed microsatellite instability (MSI) detection using immunohistochemistry (IHC) staining of four proteins (MLH1, MSH2, MSH6 and PMS2)^{8,9}. We found that only 9 of 36 samples were MSI-high samples (revised **Supplementary Figure 8a, 8b**, revised **Supplementary Table 4**), while 27 patients were MSI-low. The two chromothripsis-negative samples were both in the MSI-low group (revised **Supplementary Fig 8b**), and there was no correlation between MSI grade and chromothripsis events (revised **Supplementary Figure 8b**, $p = 1$, Fisher's exact test). Thus, we concluded that the high frequency of chromothripsis was not likely due to the high proportion of MSI-high samples in our cohort. We have included this part of the study in the revised manuscript (**Lines 278-285**, revised manuscript).

ii), We followed the same method in Birkbak et al., 2011 Cancer Research¹⁰ (PMID: 21270108) to calculate the overall pattern of chromosomal instability (CIN) for all 36 samples and divided our GCA patients into four groups based on the genome integrity index (from low to high: 0 to 0.2, 0.2 to 0.4, 0.4 to 0.6, 0.6-0.8). Only 2 samples were in the high-grade CIN group (revised **Supplementary Figure 8c**, revised **Supplementary Table 4**). The two chromothripsis-negative samples were both in the low-grade CIN groups (revised **Supplementary Figure 8c**), and there was no correlation between CIN grade and chromothripsis events (revised **Supplementary Figure 8c**, $p = 0.381$, Fisher's exact test). Thus, we concluded that the high frequency of chromothripsis is not likely due to the high proportion of high-grade CIN in our cohort. We have included this part of the study in the revised manuscript (**Lines 285-293**, revised manuscript).

Supplementary Figure 8

The detection of microsatellite instability (MSI) (revised Supplementary Figure 8a, 8b) and chromosomal instability (CIN) (revised Supplementary Figure 8c) in our GCA cohort.

iii), We performed IHC of the γ H2AX protein, a crucial biomarker for the detection of DNA double strand breaks¹¹, in our GCA cohort. We found that 80.55% (29/36) of GCA patients were γ H2AX protein positive (**Fig. 3e, 3f, Supplementary Table 4**). The two chromothripsis-negative samples were both γ H2AX protein negative (**Fig. 3f**), and there was a significant correlation between the presence of γ H2AX and chromothripsis events (**Fig. 3f**, $p = 0.033$, Fisher's exact test). We also found that the total length of chromothripsis from the genome was significantly longer (revised **Figure 3g**, $p = 0.025$, revised

Supplementary Table 4) in γ H2AX protein-positive patients than in γ H2AX protein-negative patients. Thus, we think the high frequency of chromothripsis in our GCA cohort is most likely due to GCA patients in our study being from the high incidence area for GCA in Henan Province, northern China¹², where the intake of nitrosamine-rich foods, such as pickled vegetables, has been well recognized as one of the key risk factors for GCA¹³. Accumulating evidence has demonstrated that nitrosamine is a very important factor for DNA alkylation, synthesis disorder, high instability and even DNA double strand breakage¹⁴⁻¹⁹. Thus, we suspect that nitrosamine exposure in this population may increase DNA damage and further trigger the high frequency of chromothripsis. We have included this part of the study in the revised manuscript (**Lines 293-313**, revised manuscript).

Figure 3e

Figure 3f

Figure 3g

Representative image of γ H2AX protein staining in GCA patients (revised Figure 3e). Presence and absence of chromothripsis in γ H2AX-positive and γ H2AX-negative GCA patients (revised Figure 3f). The numbers on the bars are patient numbers and the relationship of chromothripsis and chromothripsis (revised Fig. 3g, p-value was calculated using Wilcoxon signed-rank test.

Minor comments:

1. Please clarify what is meant by the following statements: “However, the frequency of these ecDNA amplicons is higher in the coding region (6.28%) compared to the whole genome (3.48%) (Supplementary Fig. 1g). Furthermore, the proportion of ecDNA amplicons detected in the exons is higher (14.5%) compared to the whole genome (9.2%) (Supplementary Fig. 1h).”

Thank you for your suggestions. Our text has been updated to address the concern of unclear descriptions. In this description, our purpose was to compare the proportion of coding sequences in the entire genome and ecDNA amplicon regions, where we found that 3.48% of the entire genome is occupied by coding regions, and 6.28% of all ecDNA amplicon regions are from coding regions. At the same time, we compared the proportion of exons in the whole genome (9.2%) and ecDNA amplicon regions (14.5%).

Here comes our updated description (**Lines 141-145**, revised Manuscript): However, the frequency of ecDNA amplicons observed in coding regions (6.28%) was higher than the proportion of coding regions in the whole genome (3.48%) (**Supplementary Figure 2f**). Furthermore, the proportion of ecDNA amplicons detected in the exons (14.5%) is higher than that of exons in the entire genome (9.2%) (**Supplementary Figure 2g**).

2. Overexpression of CCNE1 is mentioned, but there is no mention of RNA expression data for the cohort.

Thank you. That is a truly good point. Reviewer 2 pointed out the overexpression of CCNE1 (see below,

From reviewer 2, minor, Line 89. “Overexpression” is more specific to protein and should not be used to describe the amplification of a gene), we apologize that we used the wrong word “overexpression of CCNE1”, which is supported by gene amplification. We have corrected it to gene amplification in the revised manuscript.

Reviewer #2 (Remarks to the Author): Expert in cancer genomics

In this manuscript, Zhao et al. investigated the presence and the role of extrachromosomal circular DNA (ecDNA) in a cohort of 36 patients with gastric cardia adenocarcinoma. They used whole-genome sequencing of tumor and adjacent normal DNA and the AmpliconArchitech algorithm to assess the presence of ecDNA. They explored the role of ecDNA by annotating amplicons and focused their attention on oncogenes. They used ShatterSeek to explore the role of chromothripsis in the etiology of ecDNA and claimed that a third of scDNA might be explained by chromothripsis. Finally, they investigate the impact of CCNE1, EGFR and ERBB2 copy number on survival and used a larger cohort to confirm that HER2 overexpression is associated with survival.

Thank you for thoughtful comments to improve our manuscript.

The manuscript presents several major flaws.

Major:

1. Generally, the manuscript is not well written and sometimes not intelligible. The figure legends are incomplete.

Thanks for your comments. In our revised manuscript, we performed language proofreading using a professional English proofreading firm. Our figure legends have been updated and confirmed by the English proofreading firm as well.

The study design (patients' selection, follow-up time) is lacking and the methodology (WGS, ecDNA prediction) is not rigorous enough to draw any valid conclusions.

Thanks for your comments. Our detailed experimental design is included in the material and methods of the revised manuscript.

A brief summary:

i), Patient selection:

All GCA patients were from our oesophageal cancer database (years from 1973 to 2020), which was established and is maintained by the State Key Laboratory for Oesophageal Cancer Research of the First Affiliated Hospital, Zhengzhou University. All patients involved in our study were from the same hospital, and the clinicopathological measurements for all patients had the same parameters. None of the patients were treated with radiotherapy or chemotherapy before surgery.

ii), Follow-up time

A detailed description of the follow-up visit plan and strategy is included in the materials and methods of the revised manuscript. In brief, every patient was visited at home with the following frequency after surgery: once every three months during the first year, once each 6 months during the second year, and once per year after the third year.

iii), QC of WGS data

In our first submission of the manuscript, we only showed the sequencing coverage of our whole-genome sequencing data with a bar graph in Supplementary Figure S1a. In our revised manuscript, the detailed QCs of WGS data were further performed from two angles: i) we performed a detailed QC of our sequencing data, including sequencing depth, coverage, sequencing quality, and mapping rate, and have included these QC data in the **revised Supplementary Table 1**; and ii) we calculated the gene mutation frequency from WGS and compared the top mutated genes list with classical driver mutated genes present in a previous GCA study² and have included this result (**Lines 92-95**, revised manuscript) in the **revised Supplementary Figure 1b**.

Supplementary Table 1

Sample ID	Reads after removing duplicates	>Q20 (Percentage)	>Q30 (Percentage)	Mapping_rate	Coverage
S1_Tumor	33838927950	0.95	0.89	99.93%	10.91
S2_Tumor	38874477450	0.95	0.89	99.90%	12.53
S3_Tumor	32234315250	0.95	0.89	99.91%	10.39
S4_Tumor	37106553150	0.95	0.90	99.93%	11.96
S5_Tumor	79017403950	0.95	0.88	99.93%	25.47
S6_Tumor	65857136100	0.95	0.88	99.70%	21.23
S7_Tumor	65666901300	0.95	0.89	99.76%	21.17
S8_Tumor	75468588300	0.94	0.87	96.13%	24.33
S9_Tumor	69980563050	0.95	0.88	99.38%	22.56
S10_Tumor	90095794350	0.95	0.89	99.65%	29.05
S11_Tumor	86715235350	0.96	0.90	99.62%	27.96
S12_Tumor	76455384750	0.96	0.91	99.59%	24.65

Supplementary Figure 1b

QC on whole-genome sequencing data from GCA patients. Part of **Supplementary Table 1**: Detailed QC of WGS data; **Supplementary Figure 1b**: Top mutated genes identified in 36 patients of a GCA cohort using whole-genome sequencing data.

iv), ecDNA prediction

A more detailed description of ecDNA prediction is included in the materials and methods of the revised manuscript. Our computational code has been deposited on GitHub with the following link: <https://github.com/chenlab2019/ecDNA-on-GCA>

Importantly, the study is not reproducible as the original data and the code to regenerate the results are not available.

Thanks for your comments. All suggested items have been included in the materials and methods of the revised manuscript, where our raw data has been deposited in the China National Center for Bioinformatics: [https://bigd.big.ac.cn/? blank](https://bigd.big.ac.cn/?blank), access number: HRS000814, and our original computation scripts are shared in GitHub: <https://github.com/chenlab2019/ecDNA-on-GCA>

2. The method section could benefit from a more rigorous and detailed description. Crucial details on the samples used for this study is lacking. For example, was the DNA extracted from FFPE or frozen samples? This is important for subsequent analysis as DNA extracted from FFPE is often degraded by the fixation method. Were the tumor and normal samples assessed by a trained pathologist to assess tumor content and the absence of tumor cells in the normal adjacent tissues?

Thanks for your suggestions. More detailed experimental methods are included in our revised manuscript. For this particular case you indicated here. In brief, all DNA samples were extracted from snap freezing GCA tissue. None of the patients received any treatment before surgery. In our pathological scanning, the diagnosis of GCA patients was always identified by two well-trained pathologists. Haematoxylin and eosin (HE) staining was used to quantify the content of tumour cells in tissue sections. Only GCA tumours with more than 80% tumour cells were used for our study. The matched normal tissue samples were selected from the adjacent epithelial tissue 5-10 cm away from the edge of the tumour.

3. The quality of the sequencing is hard to assess. Apart from the sequencing depth (for which the method to compute is not described) present in figure S1a, no other metrics are present. For example, it would have been interesting to see if the classical driver mutations (TP53, CDH1, PIK3CA) present in GCA were also found in the present study.

Thanks for your suggestion. In our first version of the manuscript, we only showed the sequencing coverage for our whole-genome sequencing data with a bar graph in Figure S1a. In our revised manuscript, the more detailed QC data of our whole-genome sequencing are included in **Supplementary Table 1**, where the sequencing depth, coverage, sequencing quality, and mapping rate are described (see the **answer to question 1 iii**). The detailed computational workflow and code are now described in our methods section. It is a great suggestion to assess the quality of sequencing data with mutation detection, and we included this in our revised manuscript. In brief, we performed somatic SNV calling using Mutect2 tools based on GATK4 software. We annotated these SNVs using the ‘Funcotator’ tools of GATK4 software. The frequency of mutations was calculated and visualized using the maftools package²². We included one heatmap to illustrate the top 20 mutated genes (revised **Supplementary Figure 1b**). The top mutated gene was TP53, with an 81% mutation rate, in agreement with a previous report². Interestingly, the mutation rate of CDH1 and PIK3CA was not very high. In China, gastric cardia adenocarcinoma (GCA) and oesophageal squamous cell carcinoma (ESCC) occur together in the Taihang Mountains of north central China at high rates¹², and all of our samples were collected in this area. Gastric cancer in this area occurs primarily in the uppermost portion of the stomach and is referred to as gastric cardia adenocarcinoma (GCA), and those in the remainder of the stomach are called gastric noncardia adenocarcinoma (GNCA)². Thus, the low mutation frequency of the CDH1 and PIK3CA genes likely reflects the specific features of GCA from the selected geographic areas of China^{2,12,23-25}.

The mutation frequency of the TP53, PIK3CA and CDH1 genes identified from 36 pairs of GCA cohorts with whole-genome sequencing data

4. Basic clinicopathological variables of the patients included in this study are lacking (age at diagnosis, sex, grade, stage, histology, molecular classification, EBV status).

Thank you for your suggestions. In our revised manuscript, we provided more detailed clinicopathological information for all patients in revised **Supplementary Table 4**.

Among other important omissions, the specific library used for the WGS is not detailed and the WES from the independent 75 GCA patients is not described.

Detailed descriptions of the specific library for both 36 pairs of WGS and 75 WES are included in the materials and methods of the revised manuscript.

It is unclear how exactly the authors have annotated the genomic elements present in the predicted cDNA.

A more detailed description of genomic feature annotation from ecDNA amplicon regions is included in the revised manuscript. Our codes are shared on GitHub now. In brief, regions of the ecDNA amplicons were extracted from the output of AA software. For the genomic elements annotation (revised **Figure 1c**), we first performed the intersection between genomic elements from reference genome (hg19) and ecDNA region with bedtools²⁶, then we calculated the length of overlapping regions between genomic elements from reference genome and ecDNA regions, where the genomic elements were annotated to ecDNA amplicons if there was one bp or more overlapping. For the occupancy of coding regions and exons regions in ecDNA amplicons (revised **supplementary Figure 2f, 2g**), we calculated with following formulas:

$$\text{occupancy of coding regions in ecDNA (\%)} = \frac{\text{total length of coding regions in all ecDNA amplicons}}{\text{total length of all ecDNA amplicons}} \times 100$$

$$\text{occupancy of exon regions in ecDNA (\%)} = \frac{\text{total length of exon regions in all ecDNA amplicons}}{\text{total length of all ecDNA amplicons}} \times 100$$

5. The timepoints used for overall survival are not defined.

Thanks for your comments. In the revised manuscript, a detailed description of survival is provided in the materials and methods. In brief, the definition of overall survival time for dead patients is a period from diagnosis to death, and the definition of overall survival time for alive patients is a period from diagnosis to last follow-up visit (Jan 2021).

The statistic to assess the impact of ecDNA on survival is incorrect. In general, survival analysis is meaningless without taking into consideration clinicopathological variables of patients (what are the characteristics of patients with and without ecDNA?)

Thanks for your comments. In the revised manuscript, we investigated the relationship of six clinicopathological phenotypes (sex, age, cigarette smoking, alcohol consumption, family history and UICC tumour stage) and the presence of ecDNA amplicons of ERBB2, EGFR, and CCNE1. The statistical methods we chose were as follows: Fisher's exact test for sex, family history of cigarette smoking, alcohol consumption and tumour stage, and Wilcoxon signed-rank test for age. This information is all included in the revised manuscript.

First, we performed an overall survival analysis based on clinicopathological phenotypes (sex, age, cigarette smoking, alcohol consumption, family history and UICC tumour stage), and we concluded that UICC tumour stage was the only clinicopathological factor affecting GCA survival (revised **Supplementary Figure 12**).

Second, we performed survival analysis using clinicopathological variables of patients together with the presence of ecDNA amplicons (ERBB2, EGFR, CCNE1) by dividing patients into those with and without ecDNA amplicons (revised **Supplementary Figure 12**). We found that the presence of ERBB2 ecDNA

amplicons may be relevant to the UICC tumour stage but not to other clinicopathological variables. However, the presence of EGFR and CCNE1 ecDNA amplicons was not relevant to any clinicopathological variables. In short, we identified 4 patients with ERBB2 ecDNA, three of whom were in tumour stage II and one of whom was in tumour stage III. Since both UICC tumour stage and the presence of ERBB2 ecDNA are contributing factors to patient survival, we assumed that there might be some potential relationship between ecDNA amplification of ERBB2 and tumour stage. However, our sample size was too small (36 cases) to make further conclusions. We discussed this point in the Results section of our revised manuscript (**Lines 402-414**, revised manuscript). It will be interesting to perform further studies using larger sample sizes of patients to obtain better conclusions in the future, which could be independent from our current study.

Supplementary Figure 12

Survival analysis based on clinicopathological phenotypes (sex (revised **Supplementary Figure 12a**), cigarette smoking (revised **Supplementary Figure 12c**), alcohol consumption (revised **Supplementary Figure 12e**), family history (revised **Supplementary Figure 12g**) and UICC tumour stage (revised **Supplementary Figure 12i**), age (revised **Supplementary Figure 12k**)).

Supplementary Figure 12

Survival analysis with clinicopathological variables of patients together with the presence of ecDNA amplicons (ERBB2, EGFR, CCNE1) was performed by dividing patients into groups with and without ecDNA amplicons (revised Supplementary Figure 12b, revised Supplementary Figure 12d, revised Supplementary Figure 12f, revised Supplementary Figure 12h, revised Supplementary Figure 12j, revised Supplementary Figure 12l).

6. In Figure 1e and the text, the authors describe the presence of ecDNA measuring 40Mb. To put this in context the whole chromosome 21 measures 47Mb. It was previously reported that the size of ecDNA in human cancer was between 1 and 3Mb (<https://www.nature.com/articles/s41568-019-0128-6>). Could the authors comment on this big difference?

We apologize for the unclear description in the previous version, which has been updated in the revised version. The size of the largest amplicon was 22.6 Mb. The 42.8 Mb was the total amplicon length on chromosome 19. To avoid misunderstanding, we removed the summary of total ecDNA amplicon size in each chromosome (it is not very informative). In our revised manuscript, we characterized the size of ecDNA amplicons in the following format: **i**) size of ecDNA amplicons in each sample (revised **Supplementary Figure 2a**); **ii**) size distribution of ecDNA amplicons from all samples (revised **Supplementary Figure 2b**), where 75% of ecDNA amplicons are smaller than 1 Mbp, 90% of ecDNA amplicons are in the range of 0-5 Mbp, 3% of ecDNA amplicons are larger than 10 Mbp, and only 1% of ecDNA amplicons are larger than 20 Mb; **iii**) size of ecDNA amplicons on each chromosome (revised **Supplementary Figure 2d**).

Supplementary Figure 2a

Supplementary Figure 2b

Supplementary Figure 2d

Characterizing the size of ecDNA amplicons. Supplementary Figure 2a: Summary of ecDNA amplicon size in each GCA patient, where each dot represents one ecDNA amplicon; **Supplementary Figure 2b:** Size distribution of ecDNA amplicons in our GCA cohort; **Supplementary Figure 2d:** Summary of ecDNA amplicon size in each chromosome from the cohort, and each dot represents one ecDNA amplicon.

7. The reported frequency of chromothripsis event in the present study is 35/36 (97%) which is surprisingly high compared to what was reported using the same method (50%, <https://www.nature.com/articles/s41588-019-0576-7>). Could the authors comment? Did the cohort include some MSI-high/CIN tumors?

Thanks. This is a truly good point. We double checked the frequency of chromothripsis and recalculated it. We realized that we made a miscalculation, and the actual chromothripsis rate was 34/36 (94%), which

has been updated in the revised manuscript. The frequency is indeed very high compared to a previous study²⁷(50%, <https://www.nature.com/articles/s41588-019-0576-7>) as the reviewer pointed out.

In our previous version of the manuscript, we investigated the relationship between genome chromothripsis and ecDNA amplicons, and our results indicate that ecDNA amplicons in GCAs are more likely to occur due to chromothripsis. However, we did not investigate the potential risk factors for chromothripsis. In our revised manuscript, we investigated the potential reason for the high frequency of chromothripsis in our GCA cohort from three angles:

i), We performed microsatellite instability (MSI) detection using immunohistochemistry (IHC) staining of four proteins (MLH1, MSH2, MSH6 and PMS2)^{8,9}. We found that only 9 of 36 samples were MSI-high samples (revised **Supplementary Figure 8a, 8b**, revised **Supplementary Table 4**), and 27 patients were MSI-low. The two chromothripsis-negative samples were all in the MSI-low group (revised **Supplementary Figure 8b**), and there was no correlation between MSI grade and chromothripsis events (revised **Supplementary Figure 8b**, $p = 1$, Fisher's exact test). Thus, we concluded that the high frequency of chromothripsis was not likely due to the high proportion of MSI-high samples in our cohort. We have included this part of the study in the revised manuscript (**Lines 278-285**, revised manuscript).

ii), We followed the same method as Birkbak et al., 2011 Cancer Research¹⁰ (PMID: 21270108) to calculate the overall pattern of chromosomal instability (CIN) for all 36 samples and divided our GCA patients into four groups based on the genome integrity index (from low to high: 0 to 0.2, 0.2 to 0.4, 0.4 to 0.6, 0.6-0.8). Only 2 samples were in the high-grade CIN group (revised **Supplementary Figure 8c**, revised **Supplementary Table 4**). The two chromothripsis-negative samples were both in the low-grade CIN groups (revised **Supplementary Figure 8c**), and there was no correlation between CIN grade and chromothripsis events (revised **Supplementary Figure 8c**, $p = 0.381$, Fisher's exact test). Thus, we concluded that the high frequency of chromothripsis was not likely due to the high proportion of high-grade CIN in our cohort. We have included this part of the study in the revised manuscript (**Lines 285-293**, revised manuscript).

Supplementary Figure 8

The detection of microsatellite instability (MSI) (revised Supplementary Figure 8a, 8b) and chromosomal instability (CIN) (revised Supplementary Figure 8c) in our GCA cohort.

iii), We performed IHC of γ H2AX protein, a crucial biomarker for the detection of DNA double strand breaks¹¹, in our GCA cohort. We found that 80.55% (29/36) of GCA patients were γ H2AX protein positive (**Fig. 3e, 3f, Supplementary Table 4**). The two chromothripsis-negative samples were both γ H2AX protein negative (**Fig. 3f**), and there was a significant correlation between the presence of γ H2AX and chromothripsis events (**Fig. 3f**, $p = 0.033$, Fisher's exact test). We also found that the total length of chromothripsis from the genome was significantly longer (revised **Figure 3g**, $p = 0.025$, revised **Supplementary Table 4**) in γ H2AX protein-positive patients than in γ H2AX protein-negative patients. Thus, we think the high frequency of chromothripsis in our GCA cohort is most likely because GCA patients in our study are from the high incidence area for GCA in Henan Province, northern China¹², where the intake of nitrosamine-rich foods, such as pickled vegetables, has been well recognized as one of the key risk factors for GCA¹³. Accumulating evidence has demonstrated that nitrosamine is a very important factor for DNA alkylation, synthesis disorder, high instability and even DNA double strand breaks¹⁴⁻¹⁹. Thus, we suspect that nitrosamine exposure in this population may increase DNA damage and further trigger the high frequency of chromothripsis. We have included this part of the study in the revised manuscript (**Lines 293-313**, revised manuscript).

Figure 3e

Figure 3f

Figure 3g

Representative image of γ H2AX protein staining in GCA patients (revised Figure 3e). Presence and absence of chromothripsis in γ H2AX-positive and γ H2AX-negative GAC patients (revised Figure 3f). The numbers on the bars are patient numbers and the relationship of chromothripsis and chromothripsis (revised Fig. 3g, p value was calculated using the Wilcoxon signed-rank test).

We thought such a high frequency of chromothripsis in GCA patients was most likely due to the local living environment. All patients with gastric cardia adenocarcinoma (GCA) in our study were from the high incidence area for GCA in Henan Province, northern China, where the intake of nitrosamine-rich foods, such as pickled vegetables, has been well recognized as one of the key risk factors for GCA. Accumulating evidence has demonstrated that nitrosamine is a very important factor for DNA alkylation, synthesis disorder, high instability and even breakage. Thus, we speculate that nitrosamine exposure in this population may increase genomic instability, explaining the high frequency of chromothripsis and DNA breaks. To test our hypothesis, we performed IHC of γ H2AX protein expression, a crucial biomarker for the detection of DNA double strand breaks, on our 36 samples. We found that the positive IHC rate for γ H2AX protein was 80.55% (29/36), which was very high and was very similar to the

frequency of ecDNA occurrence that we reported in this study. This result has been added to our revised manuscript.

8. It is unclear if the cohort of 1668 patients used to assess the impact of HER2 amplification on survival was done in the context of this study. Does the IHC and pathological assessment were done to corroborate the results or does the data was preexisting?

Thank you for your comments. We apologize for the unclear description, which has been addressed in our revised manuscript. The cohort of 1668 patients was obtained from our esophageal cancer database (years from 1973 to 2020), which was established and is maintained by the State Key Laboratory for esophageal Cancer Research of the First Affiliated Hospital, Zhengzhou University. All patients involved in our study were from the same hospital, and the clinicopathological measurements for all patients had the same parameters. HER2 (ERBB2) IHC was newly performed in our study. We had include the detail description in our revised materials and methods.

Could the authors provide more details on how they assessed the presence/absence of HER2 amplification?

Perhaps our unclear description in the first version misled the reviewer. In the 1668-patient study, we did not assess the presence or absence of HER2 amplification. We determined the protein expression of HER2 using IHC staining and divided patients into positive and negative HER2 expression groups. The reason we did so is because **i)** high amplification of ecDNA was relevant to the high gene expression in a previous report²⁰; **ii)** It was also reported the amplification of *ERBB2* gene was followed by *ERBB2* gene overexpression in the same GCA tissue^{23,28-30}, and **iii)** we learned that there was a positive correlation (Pearson corr.= 0.79, $p < 0.001$) between gene expression and protein expression from ERBB2 from our independent study in GCAs (n = 44) (revised **Supplementary Figure 14a**).

Supplementary Figure 14a

Correlation of ERBB2 gene expression and protein expression in GCA patients (n = 44) (revised Supplementary Figure 14a).

In our revised manuscript, a detailed description and assessment of HER2 protein expression results from IHC staining are provided in the Materials and Methods.

There is a discrepancy between the number of patients referred in the text (n=1668) and the legend of figure 4d (n=1598). Furthermore, the p-value indicated in the text and figure 4d (p=0.027) is not the same as the one in figure S7 (p=0.16). Supplementary table: only survival time is present (the unit is absent, is it years, months?). The event status is not present whereas in figure 4d it seems that some patients are censored. Could the author address those issues?

Once again, we apologize for the unclear description, and we have updated the revised manuscript. Among the 1668 patients in our HER2 IHC staining study, 33 samples contained WGS data that were used in our ecDNA analysis, 37 samples contained WES data, and 1598 samples were from independent collections. The label of patient number is wrong in the previous manuscript. We apologize that the survival curve 0-2 years after surgery was not provided in the previous manuscript, which is now included in the revised manuscript (left panel revised **Figure 4d**).

Survival analysis dividing patients into positive and negative ERBB2 staining: With all patients (n = 1668) (revised **Supplementary Figure 14c**), patient prognosis was between 0-2 years (including 2 years) (n = 750) (revised **Figure 4d**), and patient prognosis was between 2-7 years (n = 918) (revised **Figure 4d**).

Since the chromatin structure of ecDNA is highly accessible²⁰, it was also reported the amplification of ERBB2 gene was followed by ERBB2 gene overexpression in the same GCA tissue^{23,28-30}, and there is a positive correlation between ERBB2 gene expression and protein expression in the GCA cohort (revised **Supplementary Figure 14a**), we assumed that the protein level of ERBB2 is high in ecDNA-positive patients and that a high level of ERBB2 protein is also positively associated with GCA prognosis. When we performed survival analysis by dividing patients into positive and negative ERBB2 staining groups, we found no significant difference in patient prognosis among all patients (n = 1668, revised **Supplementary Figure 14c**, p = 0.16), but there was a significant difference in patient prognosis between 0-2 years (including 2 years) after surgery (n = 750, left panel revised **Figure 4d**, p = 0.016) and between 2-7 years (n = 918, right panel revised **Figure 4d**, p = 0.025). Our conclusion is that there is a positive correlation between HER2 presence and patient prognosis in the 2-7 years' survival after surgery; however, there is a negative correlation between HER2 presence and patient prognosis in the 0-2 year survival after surgery. Since we assumed that the protein level of ERBB2 was high in ecDNA-positive patients, our observation indicates that *ERBB2* ecDNA may represent a prognostic marker in GCA patients with longer survival times.

9. Correlative analysis of clinical features of patients and specific molecular subtypes (EBV, MSI, CIN, GS) could strengthen the study.

Thanks, this is a great suggestion. In gastric cancer, specific molecular subtypes (EBV, MSI, CIN and GS) have been identified and well characterized³¹. Previous studies have shown that GCA and gastric cancer are different cancer types with respect to their epidemiological characteristics and molecular subtype in China^{2,12,23-25}, but there is no clear definition of specific molecular subtypes of GCA. We have added this point to the introduction of the revised manuscript.

We followed the reviewer's suggestion to perform EBV detection using in situ hybridization, MSI with IHC staining, and calculated the CIN in our cohort (Methods and revised **Supplementary Table 4**). We found **i**) that all 36 samples were EBV negative; **ii**) only 9 samples were MSI unstable; and **iii**) only two samples were CIN high grade. All of the above information is provided in the revised **Supplementary**

Table 4. We also investigated the relationship between MSI, CIN and chromothripsis in our revised manuscript.

Extrachromosomal DNA is a major mechanism of drug resistance in several tumor types (<https://www.nature.com/articles/s41586-020-03064-z>). No clinical annotation on previous exposure therapy is currently present but it would be interesting to explore this in more detail.

Thanks, this is a really great suggestion. It will be valuable to follow clinical annotation on previous exposure therapy together with ecDNA detection in large-scale samples of GCA patients for the design of therapy strategies. Our ecDNA prediction and detection in our current work are only from 36 GCA patients, and it is not easy to perform this type of studies with small sample size. We think it is very reasonable to perform this type of research with large-scale numbers of patients or xenograft mouse models in the future, and we are working on that as an independent study with our 1668 GAC cohort. We have discussed this point in our revised manuscript (**Lines 474-478**, revised manuscript).

Minor:

The introduction and discussion are not titled.

This point is well taken and has been fixed in our revised manuscript.

Figure 3d. The number of chromothripsis events goes from 0 to 20. In the text, the upper range is 14 (line 166)

Once again, we apologize for the unclear description that misled the reviewer. We have fixed the language in the revised manuscript. In Figure 3d, the largest number of chromothripsis events was 17. We divided chromothripsis into high confidence (HC) and low confidence (LC), where the range of chromothripsis events was from 0 to 4 for HC, 0 to 14 for LC, and 0 to 17 for all.

Line 33. What “diverse correlation with patients’ prognosis” means here

Thanks. ‘Diverse’ in our manuscript means that the correlation between the presence of ecDNA and prognosis is different depending on the gene. We have explained this in the revised manuscript (**Lines 391-398**, revised manuscript): The survival curve in revised **Figure 4c** clearly shows that there is no significant difference in patient prognosis for the absence or presence of CCNE1 ecDNA ($p = 0.55$); the presence of EGFR ecDNA has a negative correlation with patient prognosis (revised **Figure 4c**, $p = 0.036$); and the presence of ERBB2 ecDNA has a positive correlation with patient prognosis (revised **Figure 4c**, $p = 0.0068$).

Figure 4c

The relationship between prognosis and the presence of oncogene ecDNA (*ERBB2*, *EGFR* and *CCNE1*) (revised **Figure 4c**).

Line 44. What other “human disease” please clarify

Thanks. We have fixed this in the revised manuscript.

Line 51-52. Please explain the different features associated with GCA from Chinese pop? Citation from 1987, are there any more recent epidemiologic studies?

Thanks. We added recent epidemiologic studies as references (Reference numbers 9-19) in the revised manuscript. The primarily different features are included in the introduction of our revised manuscript (**Lines 62-75**, revised manuscript).

Line 62. CNV is related to germline variations. Please use CNAs when referring to somatic.

Thanks for your suggestion. We have used CNAs in our revised manuscript.

Line 68-71. The size of ecDNAs are reported to be from 100kb to 22.6Mb (line 68) and then from 62kb to 42.8Mb (line 71). It is very unclear if the authors describe the sum of all ecDNA per sample or the real size.

Thank you for your comments, which have been used to update the manuscript with a clearer language. We apologize for the unclear description in the previous version, which has been updated in the revised version. The size of the largest amplicon was 22.6 Mb. The 42.8 Mb was the total amplicon length on chromosome 19. To avoid misunderstanding, we removed the summary of total ecDNA amplicon size in each chromosome (it is not very informative). In our revised manuscript, we characterized the size of ecDNA amplicons in the following format: **i**) size of ecDNA amplicons in each sample (revised **Supplementary Figure 2a**); **ii**) size distribution of ecDNA amplicons from all samples (revised **Supplementary Figure 2b**), where 75% of ecDNA amplicons are smaller than 1 Mbp, 90% of ecDNA amplicons are in the range of 0-5 Mbp, 3% of ecDNA amplicons are larger than 10 Mbp, and only 1% of ecDNA amplicons are larger than 20 Mb; **iii**) size of ecDNA amplicons on each chromosome (revised **Supplementary Figure 2d**).

Suppl Figure 1f. the Barplot is not needed.

Thanks. We removed Suppl Figure 1f in the revised manuscript.

Line 89. “Overexpression” more specific to protein and should not be used to describe the amplification of a gene.

Thanks. We apologize for the incorrect word being used, and we have changed it accordingly.

Line 96. CDK12 is not an oncogene but a TSG.

Thanks. It is a great suggestion. In the previous version of our manuscript, we used the oncogene list from AmpliconArchitect (AA). We agree that CDK family genes are tumour suppressor genes. In our revised version, we have modified our text to refer to oncogenes and tumour suppressor genes (TSG).

Line 100-103. This sentence is not intelligible.

Thanks. We have updated it accordingly.

Line 119. What “computational tool” please detailed.

Thanks. We have modified it accordingly in the revised manuscript (**Lines 87-88**, revised manuscript).

Line 235. Table number is X.

Thanks, we apologized for our sloppy work. It was fixed accordingly now.

Line 399. Please provide the exact reference of the antibody used (ID, manufacturer).

Thanks. We have fixed this in the revised manuscript.

Figure 4c. There is no “low” category for the EGFR survival analysis.

Thank you for the comment. It is a truly very good point.

In the previous version of the main Figure 4b (revised **Supplementary Figure 10**), we determined low, normal and high categories of three gene amplifications based on copy number amplifications. For *CCNE1* and *ERBB2*, the distribution of amplification numbers in 36 patients exhibited a parabolic pattern, and we divided gene amplifications from these two genes into three groups: low, normal and high. However, the distribution pattern is different from the amplification of *EGFR*, and there were only two groups, high and normal. We updated our figure legends and stated it in more detail in figure legends now.

Reviewer #3 (Remarks to the Author): Expert in gastric cancer genomics and clinical research
 In this study, Zhao and colleagues report a survey of extrachromosomal DNA (ecDNA) in gastric cardia adenocarcinoma (GCA). Analyzing WGS data of 36 GCAs, they find that different patients have distinct levels of ecDNA amplicons, with varying sizes. Several of the amplicons involve known oncogenes (eg CCNE1, HER2/ERBB2), and the overall amplicon profiles are similar to gastric cancer compared to esophageal cancer. The ecDNA amplicons are enriched in certain genomic features, such as exons and protein-coding genes. Certain oncogenes (EGFR, CDK6) are found in the same ecDNA amplicon. A number of the WGS-inferred ecDNA amplicons were confirmed by circle-Seq. There is an association between specific ecDNA amplicons and regions of chromothripsis. They also report for CCNE1 and HER2 amplicons, that patients with very high levels have paradoxically improved survival.

Thank you for your thoughtful comments to improve our manuscript.

1) The finding that multiple oncogenes are found in the same ecDNA amplicon is quite interesting, as it provides an obvious mechanism for oncogene co-amplification. How often does this type of pairwise interaction occur?

Thanks. It is a great suggestion. In our revised manuscript, we investigated oncogene co-amplifications from the list of oncogene and tumour suppressor gene ecDNA amplicons (Lines 182-207, revised manuscript). We found **i)** co-amplification of ecDNA oncogenes (two or more oncogenes and/or tumour suppressor genes in the same ecDNA amplicon) occurs in 50% of patients (18 of 36 patients) (revised **Figure 1e**); **ii)** the frequency of oncogene coamplification varies from 50% to 100% among all oncogene amplifications in different patients (revised **Figure 1e**, revised **Supplementary Figure 4a**); and **iii)** some pairs of oncogene coamplification events were observed in more than one patient (revised **Supplementary Table 3**), where pairs of *ERBB2* and *CDK12*, *RARA* and *SMARCE1*, pairs of *CBLC* and *BCL3* occurred in 3 patients; pairs of *EGFR* and *IRF4*, *PPARG* and *RAF1*; and *CDK12*, *ERBB2* and *RARA* occurred in 2 patients.

Figure 1e

Supplementary Figure 4a

Oncogene co-amplification in the same ecDNA: summary from all samples (revised **Figure 1e**) and summary from individual ecDNA amplicons (revised **Supplementary Figure 4a**).

2) Can the authors investigate if there is any relationship between gene mutations and/or somatic copy number changes in the ‘regular’ nuclear genome and the presence of ecDNAs? For example, do patients with HER2 ecDNA amplicons also exhibit HER2 regular copy number amplifications?

Thanks. These points are all good suggestions.

There are some technical challenges to addressing this point in the field now. From the computational method, we know the CNAs are from regular nuclear genome in the ecDNA absent samples; however, we could not distinguish CNAs from ‘regular’ nuclear genome or from extrachromosomal gene amplification in the ecDNA present samples. From the experimental method, our study materials are from clinical patients, and there are very few cells in metaphase. Thus, it was also difficult for us to distinguish gene amplifications from regular nuclear genome or ecDNA in the ecDNA present samples. In the future, we believe we may address this point using fast-growing new technologies in this field.

With an alternative strategy, we investigated the relationship **i**) between gene mutations (SNVs) and the presence of ecDNAs (**Lines 315-332**, revised manuscript) and **ii**) between CNAs and the presence of ecDNAs.

i), Relationship of gene mutations (SNVs) and the presence of ecDNA

We quantified the number of SNVs within the regions of ecDNA amplicons and the entire genome and compared the number of SNVs from ecDNA present regions and the entire genome using normalization. We found that the number of SNVs observed in ecDNA-present regions was in the same range as that in the entire genome ($P=0.18$, revised **Supplementary Figure 9a, 9b**). We also compared the numbers of SNVs in regions of individual oncogene or tumour suppressor gene regions (same oncogene ecDNA observed in 2 or more patients) between the presence and absence of oncogene ecDNA and found that there were significantly more SNVs in the ecDNA present group only with respect to the BIRC3 gene ($p=0.031$, revised **Supplementary Figure 9c**) but not with respect to other high frequency observed oncogenes or tumour suppressor genes. Our conclusion is that there is likely no relationship between gene mutations and the presence of ecDNA amplicons.

Supplementary Figure 9

Gene mutation (SNVs) and presence of ecDNAs: Individual sample level (revised **Supplementary Figure 9a**); Number of SNVs from ecDNA regions vs the whole genome from the whole cohort (revised **Supplementary Figure 9b**).

ii), Relationship of somatic copy number changes (CNAs) and the presence of ecDNA

We compared the copy number of CNAs between the presence and absence of ecDNA in the regions of oncogene or tumour suppressor gene ecDNAs (same oncogene ecDNA observed in 2 or more patients), and we found that there were significantly more gene copy number amplifications in the preceding

ecDNA group from most oncogene regions, except CBLC, CDK12 and IRF4 (see below). Since the ecDNA detection in our study is based on CNAs, it is not surprising to observe such a difference, and we did not include this result in our revised manuscript.

Relationship of copy number changes (CNAs) and the presence of oncogene ecDNA amplicons.

3) Have the authors considered that the relationship between survival and HER2 ecDNA may relate to overall patterns of genomic instability, as reported in earlier studies such as Birkbak et al., 2011 Cancer Research (PMID: 21270108).

This is a great suggestion. We have addressed this point in our revised manuscript by investigating the relationship among survival, chromosomal instability and the presence of ERBB2 (HER2) ecDNA (**Lines 418-430**, revised manuscript).

First, we followed the same method as in Birkbak et al., 2011 Cancer Research¹⁰ (PMID: 21270108) to calculate the overall pattern of chromosomal instability (CIN) in all 36 samples and divided our GCA patients into four groups (0 to 0.2, 0.2 to 0.4, 0.4 to 0.6, 0.6-0.8). The survival from these four groups showed that there was a significant difference among the four groups (revised **Supplementary Figure**

13a). This suggests that patients with stable chromosomes may survive longer than those with unstable chromosomes.

Second, we investigated whether survival and the presence of ERBB2 ecDNA were related to overall patterns of genomic instability. We did not observe that *ERBB2* ecDNA-present samples were only enriched in specific CIN groups (revised **Supplementary Figure 13b**), and we did not find a significant difference in CIN values between *ERBB2* ecDNA-present samples and *ERBB2* ecDNA-absent samples (revised **Supplementary Figure 13c**). Thus, we concluded that the paradoxical relationship between the presence of ERBB2 ecDNA and survival outcome is independent of CIN. Such a correlation may be due to the high frequency of DNA damage triggered by the local dietary habit. We have included this part in our revised manuscript.

Supplementary Figure 13

Genomic instability and ERBB2 ecDNA: patient survival and genomic instability (revised **Supplementary Figure 13a**); the presence of ERBB2 ecDNA, copy number of ERBB2 and genomic instability (revised **Supplementary Figure 13b**); and the presence of ecDNA and genomic instability (revised **Supplementary Figure 13c**).

References:

- 1 Kim, H. *et al.* Extrachromosomal DNA is associated with oncogene amplification and poor outcome across multiple cancers. *Nat Genet* **52**, 891-+, doi:10.1038/s41588-020-0678-2 (2020).
- 2 Hu, N. *et al.* Genomic Landscape of Somatic Alterations in Esophageal Squamous Cell Carcinoma and Gastric Cancer. *Cancer Res* **76**, 1714-1723, doi:10.1158/0008-5472.Can-15-0338 (2016).
- 3 Frankell, A. M. *et al.* The landscape of selection in 551 esophageal adenocarcinomas defines genomic biomarkers for the clinic. *Nat Genet* **51**, 506-516, doi:10.1038/s41588-018-0331-5 (2019).
- 4 Suh, Y. S. *et al.* Comprehensive Molecular Characterization of Adenocarcinoma of the Gastroesophageal Junction Between Esophageal and Gastric Adenocarcinomas. *Ann Surg*, doi:10.1097/SLA.0000000000004303 (2020).
- 5 Kumar, P. *et al.* Normal and Cancerous Tissues Release Extrachromosomal Circular DNA (eccDNA) into the Circulation. *Mol Cancer Res* **15**, 1197-1205, doi:10.1158/1541-7786.Mcr-17-0095 (2017).
- 6 Dillon, L. W. *et al.* Production of Extrachromosomal MicroDNAs Is Linked to Mismatch Repair Pathways and Transcriptional Activity. *Cell Rep* **11**, 1749-1759, doi:10.1016/j.celrep.2015.05.020 (2015).
- 7 Shibata, Y. *et al.* Extrachromosomal MicroDNAs and Chromosomal Microdeletions in Normal Tissues. *Science* **336**, 82-86, doi:10.1126/science.1213307 (2012).
- 8 Lindor, N. M. *et al.* Immunohistochemistry versus microsatellite instability testing in phenotyping colorectal tumors. *J Clin Oncol* **20**, 1043-1048, doi:10.1200/JCO.2002.20.4.1043 (2002).
- 9 Chen, L. Z., Chen, G., Zheng, X. W. & Chen, Y. Expression status of four mismatch repair proteins in patients with colorectal cancer: clinical significance in 1238 cases. *Int J Clin Exp Pathol* **12**, 3685-3699 (2019).
- 10 Birkbak, N. J. *et al.* Paradoxical Relationship between Chromosomal Instability and Survival Outcome in Cancer. *Cancer Res* **71**, 3447-3452, doi:10.1158/0008-5472.Can-10-3667 (2011).
- 11 Turinetto, V. & Giachino, C. Multiple facets of histone variant H2AX: a DNA double-strand-break marker with several biological functions. *Nucleic Acids Res* **43**, 2489-2498, doi:10.1093/nar/gkv061 (2015).
- 12 Li, K. Mortality and incidence trends from esophagus cancer in selected geographic areas of china circa 1970-90. *Int J Cancer* **102**, 271-274, doi:10.1002/jhc.10706 (2002).
- 13 Taylor, P. R. *et al.* Prevention of Esophageal Cancer - the Nutrition Intervention Trials in Linxian, China. *Cancer Res* **54**, S2029-S2031 (1994).
- 14 Weitberg, A. B. & Corvese, D. Effect of vitamin E and beta-carotene on DNA strand breakage induced by tobacco-specific nitrosamines and stimulated human phagocytes. *J Exp Clin Cancer Res* **16**, 11-14 (1997).
- 15 Wang, L. *et al.* Mutations of O6-methylguanine-DNA methyltransferase gene in esophageal cancer tissues from Northern China. *Int J Cancer* **71**, 719-723, doi:10.1002/(sici)1097-0215(19970529)71:5<719::aid-ijc5>3.0.co;2-u (1997).
- 16 Deng, C. *et al.* Genetic polymorphism of human O6-alkylguanine-DNA alkyltransferase: identification of a missense variation in the active site region. *Pharmacogenetics* **9**, 81-87, doi:10.1097/00008571-199902000-00011 (1999).
- 17 Groenen, P. J. & Busink, E. Alkylating Activity in Food-Products - Especially Sauerkraut and Sour Fermented Dairy-Products after Incubation with Nitrite under Quasi-Gastric Conditions. *Food Chem Toxicol* **26**, 215-225, doi:10.1016/0278-6915(88)90122-6 (1988).
- 18 Duell, E. J. *et al.* Vitamin C transporter gene (SLC23A1 and SLC23A2) polymorphisms, plasma vitamin C levels, and gastric cancer risk in the EPIC cohort. *Genes Nutr* **8**, 549-560, doi:10.1007/s12263-013-0346-6 (2013).

- 19 Hodgson, R. M., Wiessler, M. & Kleihues, P. Preferential methylation of target organ DNA by the oesophageal carcinogen N-nitrosomethylbenzylamine. *Carcinogenesis* **1**, 861-866, doi:10.1093/carcin/1.10.861 (1980).
- 20 Wu, S. *et al.* Circular ecDNA promotes accessible chromatin and high oncogene expression. *Nature* **575**, 699-703, doi:10.1038/s41586-019-1763-5 (2019).
- 21 Deshpande, V. *et al.* Exploring the landscape of focal amplifications in cancer using AmpliconArchitect. *Nat Commun* **10**, 392, doi:10.1038/s41467-018-08200-y (2019).
- 22 Mayakonda, A., Lin, D. C., Assenov, Y., Plass, C. & Koeffler, H. P. Maftools: efficient and comprehensive analysis of somatic variants in cancer. *Genome Res* **28**, 1747-1756, doi:10.1101/gr.239244.118 (2018).
- 23 Wang, L. D., Zheng, S., Zheng, Z. Y. & Casson, A. G. Primary adenocarcinomas of lower esophagus, esophagogastric junction and gastric cardia: in special reference to China. *World J Gastroenterol* **9**, 1156-1164, doi:10.3748/wjg.v9.i6.1156 (2003).
- 24 Y. Guanrei, a. Q. S. Incidence Rate of Adenocarcinoma of the Gastric Cardia, and Endoscopic Classification of Early Cardial Carcinoma in Henan Province, the People's Republic of China. *Endoscopy* **19**, 7-10 (1987).
- 25 Wang, L. D., Zhou, Q. & Yang, C. S. Esophageal and gastric cardia epithelial cell proliferation in northern Chinese subjects living in a high-incidence area. *J Cell Biochem Suppl* **28-29**, 159-165 (1997).
- 26 Quinlan, A. R. & Hall, I. M. BEDTools: a flexible suite of utilities for comparing genomic features. *Bioinformatics* **26**, 841-842, doi:10.1093/bioinformatics/btq033 (2010).
- 27 Cortes-Ciriano, I. *et al.* Comprehensive analysis of chromothripsis in 2,658 human cancers using whole-genome sequencing. *Cancer Res* **78**, doi:10.1158/1538-7445.Am2018-Lb-378 (2018).
- 28 Park, J. B., Rhim, J. S., Park, S. C., Kimm, S. W. & Kraus, M. H. Amplification, Overexpression, and Rearrangement of the Erbb-2 Protooncogene in Primary Human Stomach Carcinomas. *Cancer Res* **49**, 6605-6609 (1989).
- 29 Huang, J. X. *et al.* HER2 gene amplification in esophageal squamous cell carcinoma is less than in gastroesophageal junction and gastric adenocarcinoma. *Oncol Lett* **6**, 13-18, doi:10.3892/ol.2013.1348 (2013).
- 30 Houldsworth, J., Cordoncardo, C., Ladanyi, M., Kelsen, D. P. & Chaganti, R. S. K. Gene Amplification in Gastric and Esophageal Adenocarcinomas. *Cancer Res* **50**, 6417-6422 (1990).
- 31 Sohn, B. H. *et al.* Clinical Significance of Four Molecular Subtypes of Gastric Cancer Identified by The Cancer Genome Atlas Project. *Clin Cancer Res*, doi:10.1158/1078-0432.CCR-16-2211 (2017).

REVIEWER COMMENTS

Reviewer #1 (Remarks to the Author):

In this revision, the authors have addressed most issues raised in the initial review. However, several new issues need to be resolved to ensure the results are accurate and communicated as such.

Major comments:

1. Showing the mutational status of cancer driver genes in Supp Fig 1B is much appreciated; the list of genes includes most of the largest genes in the genome, which are prone to acquiring mutations due to their large coding regions. Consider replacing this set of genes with a curated list of pan-cancer drivers from Lawrence et al: PMID 23770567; or another better alternative.
2. Ten cases were selected for CircleSeq validation sequencing. In seven of the ten cases, WGS predicted a circular amplicon. CircleSeq identified circular amplicons on all ten cases. As has been shown in most detail by Koche et al (PMID 31844324), two types of circular DNA elements exist in cancer: 1. Gene or regulatory element containing ecDNAs, typically >50kb and 2. Non-gene or regulatory element containing eccDNA, typically <10kb. It is important to separate these two classes, as the first class likely has a cancer-driving function where the second type of circular elements is widespread and found on many normal cell types. Please clarify what type of circular elements were detected in the three cases in which WGS did not predict a circular DNA element.
3. Per the figure from Kim et al, included by the authors in the rebuttal letter, the frequency of circular DNA amplifications in esophageal respectively gastric cancer is approximately 40% and 25%. This is considerably lower than what is reported here, in gastric cardia adenocarcinoma, and incorrectly states in the manuscript as 80% and 50%. The text should be revised to reflect this.
4. Re-reading the text, it seems that the authors used the term 'ecDNA amplicons' to reflect every amplicon that was detected: "ecDNA amplicons were further classified into five categories (Fig. 1b, Supplementary Fig. 1c-e, Supplementary Table 2): Circular (n = 45), Complex (n = 21), Linear (n = 50), breakage fusion-bridge (BFB) (n = 4) and Invalid (n = 31)". Whereas 'ecDNA amplicon' would specifically refer to circular, extrachromosomal DNA amplicons, a.k.a. the Circular category from Kim et al. The 'Complex', 'Linear' etc categories are not ecDNA amplicons. The text needs to be revised to reflect this. Furthermore, throughout the paper, any analysis related to 'ecDNA amplicons' should be limited to amplicons classified as Circular.
5. Genomic amplification is a mechanism for gene activation. Amplification of tumor suppressor genes would result in tumor regression. To annotate amplicons for the presence of tumor suppressor genes hardly makes sense. The authors need to clarify how amplicons were annotated as contained either oncogenes or TSGs.

Reviewer #2 (Remarks to the Author):

In this revised manuscript, the authors have addressed most of my previous comments except one. How the author selected the 2-7y time interval to affirm that HER2 overexpression is associated with better survival? If the two KM survival curves cross, then this is clear departure from proportional hazards. Generally, the log rank test should not be used.

My last (very minor) recommendation is that when the authors evaluate the association between scDNA or ERBB2 amplification with patient's prognosis they could rephrase the term "correlations (positive or negative)".

Reviewer #3 (Remarks to the Author):

The authors have addressed my concerns in the original review.

Overview

We thank the Reviewers for their enthusiastic assessment of our manuscript and thoughtful comments to further improve our manuscript. Following suggestions from Reviewers, we added the following main points to our revised manuscript:

- 1**, We replaced the mutation associated genes with pan cancer driver-genes list from suggested references (1-3) and updated the top mutation rate with those pan cancer driver-genes found in our GCA cohort (revised **Supplementary Figure 1b**).
- 2**, We used focal amplifications to reflect amplicons predicted by AmpliconArchitect (AA), and use ecDNA to only describe the circular category of amplicons. We changed our research theme to focal amplifications and updated our text accordingly.
- 3**, We separated circular DNA amplicons from Circle-seq data into ecDNA and eccDNA (revised **Figure 2c**).
- 4**, We added the detail protocol for oncogene annotations of focal amplifications in the revised methods.
- 5**, We removed the biased selected 0-2, and 2-7 years surviving analysis from the ERBB2 IHC staining and adjusted the description of our observation.
- 6**, We had changed our statistic model in the surviving analysis from Log rank test to Rényi test (revised **Supplementary Figure 14c**), and updated our text accordingly.

We answer reviewer comments point-by-point below.

Reviewer #1 (Remarks to the Author):

In this revision, the authors have addressed most issues raised in the initial review. However, several new issues need to be resolved to ensure the results are accurate and communicated as such.

Thank you for your positive feedback and thoughtful comments to further improve our manuscript. We answer your comments one by one with following orders: question 1, question 4, questions 2, question 3, questions 5.

Major comments:

1. Showing the mutational status of cancer driver genes in Supp Fig 1B is much appreciated; the list of genes includes most of the largest genes in the genome, which are prone to acquiring mutations due to their large coding regions. Consider replacing this set of genes with a curated list of pan-cancer drivers from Lawrence et al: PMID 23770567; or another better alternative.

Thank you for your suggestion.

It is a great suggestion to replace gene mutation list with a list of pan-cancer driver genes. In our revised manuscript, we had extracted 666 cancer driver-genes from three main references (1-3) (Lawrence et al: PMID 23770567; MH Bailey et al: PMID 29625053 and Z Sondka et al: PMID 30293088), calculated mutation rates from these 666 cancer driver-genes, and listed top 20 mutated cancer driver-genes in our GCAs cohort (revised **Supplementary Figure 1b**). We found the top 1 mutated gene is TP53 in our cohort.

Supplementary Figure 1b

Supplementary Figure 1b, Mutation frequency of cancer driver-genes from 36 (S1-S36) GCA patients.

4. Re-reading the text, it seems that the authors used the term 'ecDNA amplicons' to reflect every amplicon that was detected: "ecDNA amplicons were further classified into five categories (Fig. 1b, Supplementary Fig. 1c-e, Supplementary Table 2): Circular (n = 45), Complex (n = 21), Linear (n = 50), breakage fusion-bridge (BFB) (n = 4) and Invalid (n = 31)". Whereas 'ecDNA amplicon' would specifically refer to circular, extrachromosomal DNA amplicons, a.k.a. the Circular category from Kim et al. The 'Complex', 'Linear' etc categories

are not ecDNA amplicons. The text needs to be revised to reflect this. Furthermore, throughout the paper, any analysis related to 'ecDNA amplicons' should be limited to amplicons classified as Circular.

Thank you for your comments to further improve our statements.

EcDNA is a fast-growing and exciting field now, and the concept of ecDNA was not well defined in different studies. We apologized: Our statements of ecDNA concept in previous two versions were not accurate, where we used ecDNA to reflect all focal amplifications predicted by AmpliconArchitect (AA)(4).

In our 1st version of manuscript, we used AmpliconArchitect (AA)(4) to get the list of focal amplifications and termed all focal amplifications as ecDNA amplicons. The reason is that it was found that focal amplifications in nearly half of the samples across a variety of cancer types can be explained by circular, extrachromosomal DNA (ecDNA) formation(4, 5). However, we agreed the way we had stated was not accurate.

In the 2nd submission, we used the AmpliconArchitect-derived breakpoint graph(6) to further classify focal amplicons from AA into four categories (Circular amplification, BFB, Complex and linear), but we still termed all focal amplifications as ecDNA, which was not accurate either.

We agreed we should not use ecDNA amplicons to reflect every focal amplification, and we should use focal amplifications instead to term all amplicons predicted by AA. In our revised manuscript, ecDNA is only used for the circular category of focal amplifications. Thus, we believe it is reasonable to change our research theme to focal amplifications in our revised manuscript, and we had updated our text and further analysis accordingly.

2. Ten cases were selected for CircleSeq validation sequencing. In seven of the ten cases, WGS predicted a circular amplicon. CircleSeq identified circular amplicons on all ten cases. As has been shown in most detail by Koche et al (PMID 31844324), two types of circular DNA elements exist in cancer: 1. Gene or regulatory element containing ecDNAs, typically >50kb and 2. Non-gene or regulatory element containing eccDNA, typically <10kb. It is important to separate these two classes, as the first class likely has a cancer-driving function where the second type of circular elements is widespread and found on many normal cell types. Please clarify what type of circular elements were detected in the three cases in which WGS did not predict a circular DNA element.

Figure 2

Figure 2: a, Summary of ecDNA overlapping lists from the prediction of AmpliconArchitect (AA) and identification using Circle-seq. The y-axis is the ecDNA amplicon number from WGS prediction. Overlap: the ecDNA amplicons were identified using both AA software from WGS and Circle-Seq. None: the ecDNA amplicons were only identified using AA software but not using Circle-Seq. **c**, Circular DNA elements identified from Circle-Seq were separated into ecDNA (copy number ≥ 7) and extrachromosomal circular DNA (eccDNA) (copy number < 7).

Thank you for your suggestions.

As we had stated above (answer of your question 4), our ecDNA concept was used wrongly in our previous version, and ecDNA is only used for the circular category in our revised manuscript now. Thus, our finding was updated accordingly in the revised manuscript. Ten cases were selected for Circle-Seq validation sequencing. In four of the ten cases, WGS predicted circular amplicons (revised **Figure 2a**), and Circle-Seq identified circular amplicons on all ten cases. We further separated the list of circular DNA elements from Circle-seq data into ecDNA (copy number ≥ 7) and extrachromosomal circular DNA (eccDNA) (copy number < 7) following previous report (7) (revised **Figure 2c**). We found that all circular DNA elements from Circle-Seq in the four of six cases (S21, S28, S29, S32), where WGS did not predict focal amplifications, are only from eccDNA (all shorter than 50 kbp) (revised **Figure 2c**). However, circular DNA elements from Circle-Seq in the other two of six cases (S27, S33), where WGS did not predict focal amplifications either, contain both ecDNA and eccDNA (revised **Figure 2c**).

3. Per the figure from Kim et al, included by the authors in the rebuttal letter, the frequency of circular DNA amplifications in esophageal respectively gastric cancer is approximately 40% and 25%. This is considerably lower than what is reported here, in gastric cardia adenocarcinoma, and incorrectly states in the manuscript as 80% and 50%. The text should be revised to reflect this.

Thanks for your comments. Our unclear description of focal amplification and ecDNA in the previous version of manuscript could mislead reviewer. We revised our text in the revised manuscript. In our previous manuscript, the 80% and 50% detected in esophageal and gastric cardia adenocarcinoma(6) are from all focal amplifications. The approximately 40% and 25% are for the frequency of circular ones (ecDNA). In our GCA cohort (with 36 patients), focal amplifications were detected in 28 samples (78%), and circular ones (ecDNA) are found in 19 samples (53%).

4. Re-reading the text, it seems that the authors used the term 'ecDNA amplicons' to reflect every amplicon that was detected: "ecDNA amplicons were further classified into five categories⁷ (Fig. 1b, Supplementary Fig. 1c-e, Supplementary Table 2): Circular (n = 45), Complex (n = 21), Linear (n = 50), breakage fusion-bridge (BFB) (n = 4) and Invalid (n = 31)". Whereas 'ecDNA amplicon' would specifically refer to circular, extrachromosomal DNA amplicons, a.k.a. the Circular category from Kim et al. The 'Complex', 'Linear' etc categories are not ecDNA amplicons. The text needs to be revised to reflect this. Furthermore, throughout the paper, any analysis related to 'ecDNA amplicons' should be limited to amplicons classified as Circular.

Thanks. We answered question 4 after question 1 (see above).

5. Genomic amplification is a mechanism for gene activation. Amplification of tumor suppressor genes would result in tumor regression. To annotate amplicons for the presence of tumor suppressor genes hardly makes sense. The authors need to clarify how amplicons were annotated as contained either oncogenes or TSGs.

Thank you for your comments. We addressed your comments one by one as follows:

i. Tumor suppressor genes in the focal amplifications.

Indeed, the emphasis of our study is oncogenes in the focal amplifications, and the oncogene list in our study was from canonical oncogenes given by AmpliconArchitect (AA) software (4). There was one minor question from reviewer 2 in the 1st round of revision, where the reviewer suggested: *CDK12* is a tumor suppressor gene but not an oncogene. We believed it is a great suggestion and agreed the reviewer is correct. To make our statement more accurate, we had changed our statement from oncogenes (1st submission) to oncogene and tumor suppressor genes (TSG) (2nd submission). We learnt that *CDK12* has the oncogenic properties as well (8). Thus, we believe it is reasonable to update our statement by using oncogenes but with a note for *CDK12* in our newly revised manuscript. We updated our text (revised **Line 148-149**) as follows: Interestingly, we found that 82 focal amplifications contained oncogenes (*CDK12* was reported as a tumor suppressor gene but with oncogenic properties (8)) (**Fig. 1c**).

ii. **Oncogenes annotation.**

The detail of oncogene annotation is included in our revised methods. In brief, oncogene annotation was performed with AmpliconArchitect (AA) software (4) and AmpliconClassifier(6) (<https://github.com/jluebeck/AmpliconClassifier>), where graph and cycles files generated by AA were taken by AmpliconClassifier. A table indicating which genes are present on the focal amplifications regions was generated from AmpliconClassifier, and the list of canonical oncogenes (1-3) from the gene table were chosen as the oncogenes list in the focal amplifications. The full oncogenes or truncated oncogenes presenting on the focal amplification regions was checked by intersection between genomic coordinates of oncogenes and genomic interval of focal amplification with bedtools. In our GCA cohort, it showed 85% of oncogenes (98 of 115 oncogenes) listed by AmpliconClassifier were fully carried in the focal amplifications, and only 17 of 115 oncogenes are with truncated - 5' or 3' end in the focal amplification (revised **Supplementary Figure 3a**).

Supplementary Figure 3a

Supplementary Figure 3a, Characterisation of oncogenes carried in the focal amplifications. Top: number of oncogenes; bottom: length of oncogenes (%) carried in the focal amplifications.

Reviewer #2 (Remarks to the Author):

In this revised manuscript, the authors have addressed most of my previous comments except one. How the author selected the 2-7y time interval to affirm that HER2 overexpression is associated with better survival? If the two KM survival curves cross, then this is clear departure from proportional hazards. Generally, the log rank test should not be used.

Thank you for your positive feedback and thoughtful comments to further improve our manuscript. The longest surviving follow-up visiting of patients in our GCA cohort (1668 patients) is 7 years so far. It is a great comment about our statistic model and time interval selection. We addressed your comments one by one as follows:

- i) We apologized for the biased-selection on the truncated time point (2-7 years) for survival time analysis. In our KM survival analysis, we observed two KM curves crossed at the time point of around 2 years, then we selected the 2-7 years time interval to interpret our observation with ERBB2 protein staining. We agreed the strategy is biased, which is not the correct way. In our revised manuscript, we had removed this part of result and only described our observation. We also turned down the tone that ERBB2 overexpression is correlated with better survival. We believe it is reasonable to allow readers to judge themselves instead of drawing strong conclusion with our biased selection of truncated survival data.
- ii) Thank you for pointing out the mistake we made for the wrong statistic model. We did not realize we should not use this model if the two KM curves cross. We really appreciated the opportunity to correct the mistake in our previous version of manuscript.

In our revised manuscript, we used Rényi test following literatures (9, 10). With the new test, we found there is a significant difference of surviving probabilities ($p = 0.024$, Rényi test with Fleming Harrington function ($p=1, q=1$)) between ERBB2 positive and negative groups (revised **Supplementary Figure14c**). We updated our result as follows: To test our hypothesis, we performed immunohistochemistry of the ERBB2 protein from 1668 GCA patients (with a GCA cohort of 0- to 7- year survival time after surgery) (see **Methods, Supplementary Fig. 14b, Supplementary Table 8**). We found there is significant difference of surviving probabilities in ERBB2 positive and negative patients ($p = 0.024$, Rényi test with Fleming Harrington function ($p=1, q=1$)) (revised **Supplementary Figure14c**), where the survival probability of ERBB2 positive patients was lower than that of ERBB2 negative patients when their surviving time is under 2 years, however, the tendency became opposite when their surviving time is longer than 2 years. It was reported ERBB2 protein expression and gene amplification correlate with better survival in esophageal adenocarcinoma(11). Our observation that the survival probability of ERBB2 positive patients (when their surviving time is longer than 2 years) was longer than that of ERBB2 negative patients in our GCA cohort, probably also reflects the similarity between esophageal adenocarcinoma features and GCA. Since we assumed that the protein level of ERBB2 was high in *ERBB2* focal amplification positive patients, our observation indicates that the *ERBB2* focal amplifications probably represent a good prognostic marker in GCA patients with surviving time longer than 2 years.

Supplementary Figure 14c

Supplementary Figure 14c, Survival analysis of positive and negative ERBB2 IHC staining groups in 1668 GCA patients. The p-value was calculated using the Rényi test (with Fleming Harrington ($p = 1, q = 1$)).

My last (very minor) recommendation is that when the authors evaluate the association between scDNA or ERBB2 amplification with patient's prognosis they could rephrase the term "correlations (positive or negative)".

Thank you for your comments. We changed our text accordingly in our revised manuscript.

Reviewer #3 (Remarks to the Author):

The authors have addressed my concerns in the original review.

Thank you for your positive feedback.

References:

1. M. S. Lawrence *et al.*, Mutational heterogeneity in cancer and the search for new cancer-associated genes. *Nature* **499**, 214-218 (2013).
2. M. H. Bailey *et al.*, Comprehensive Characterization of Cancer Driver Genes and Mutations. *Cell* **174**, 1034-1035 (2018).
3. Z. Sondka *et al.*, The COSMIC Cancer Gene Census: describing genetic dysfunction across all human cancers. *Nat Rev Cancer* **18**, 696-705 (2018).
4. V. Deshpande *et al.*, Exploring the landscape of focal amplifications in cancer using AmpliconArchitect. *Nat Commun* **10**, 392 (2019).
5. K. M. Turner *et al.*, Extrachromosomal oncogene amplification drives tumour evolution and genetic heterogeneity. *Nature* **543**, 122-125 (2017).
6. H. Kim *et al.*, Extrachromosomal DNA is associated with oncogene amplification and poor outcome across multiple cancers. *Nat Genet* **52**, 891-897 (2020).
7. R. P. Koche *et al.*, Publisher Correction: Extrachromosomal circular DNA drives oncogenic genome remodeling in neuroblastoma. *Nat Genet* **52**, 464 (2020).
8. H. Paculova, J. Kohoutek, The emerging roles of CDK12 in tumorigenesis. *Cell Div* **12**, 7 (2017).

9. H. Li, D. Han, Y. Hou, H. Chen, Z. Chen, Statistical inference methods for two crossing survival curves: a comparison of methods. *PLoS One* **10**, e0116774 (2015).
10. M. Davis, and Sharon X. Xie, Caution: hazards crossing! Using the Renyi test statistic in survival analysis. *Pharma SUG* **7** (2011).
11. P. S. Plum *et al.*, HER2/neu (ERBB2) expression and gene amplification correlates with better survival in esophageal adenocarcinoma. *BMC Cancer* **19**, 38 (2019).

REVIEWERS' COMMENTS

Reviewer #1 (Remarks to the Author):

The authors have adequately addressed the comments on their revision. I support publication of this work.

Reviewer #2 (Remarks to the Author):

The authors have addressed my concerns in the second review.

Overview

We thank the Reviewers for positive feedback our manuscript.

Reviewer #1 (Remarks to the Author):

The authors have adequately addressed the comments on their revision. I support publication of this work.

Thank you for your positive feedback.

Reviewer #2 (Remarks to the Author):

The authors have addressed my concerns in the second review.

Thank you for your positive feedback.